# Boundary curvature guided programmable shape-morphing kirigami sheets

Yaoye Hong [1], Yinding Chi[1], Shuang Wu[1], Yanbin Li[1], Yong Zhu[1] & Jie Yin [1✉]

Kirigami, a traditional paper cutting art, offers a promising strategy for 2D-to-3D shape morphing through cut-guided deformation. Existing kirigami designs for target 3D curved shapes rely on intricate cut patterns in thin sheets, making the inverse design challenging. Motivated by the Gauss-Bonnet theorem that correlates the geodesic curvature along the boundary with the Gaussian curvature, here, we exploit programming the curvature of cut boundaries rather than the complex cut patterns in kirigami sheets for target 3D curved morphologies through both forward and inverse designs. The strategy largely simplifies the inverse design. Leveraging this strategy, we demonstrate its potential applications as a universal and nondestructive gripper for delicate objects, including live fish, raw egg yolk, and a human hair, as well as dynamically conformable heaters for human knees. This study opens a new avenue to encode boundary curvatures for shape-programing materials with potential applications in soft robotics and wearable devices.

[1] Department of Mechanical and Aerospace Engineering, North Carolina State University, Raleigh, NC 27695, USA. ✉email: jyin8@ncsu.edu

Designing shape-programming materials from 2D thin sheets to 3D shapes has attracted broad and increasing interest in the past decades due to their novel materials properties imparted by geometrical shapes[1]. Programmable shape shifting in different materials and structures was realized at all scales utilizing folding, bending, and buckling[2]. These shape-programmable materials are attractive for broad applications in programmable machines and robots[3,4], functional biomedical devices[5], and four-dimensional (4D) printing[6,7].

Kirigami, the traditional art of paper cutting, has recently emerged as a new promising approach for creating shape morphing structures and materials[8–19]. Cuts divide the original continuous thin sheets into discretized cut units without sacrificing the global structural integrity. Compared to continuous thin sheets, the kirigami sheet enables more freedom and flexibility in shape shifting through local or global deformation between cut units[17]. Starting from a thin sheet with patterned cuts, it can morph into varieties of 2D and pop-up 3D structures via rigid rotation mechanism[20] and/or out-of-plane buckling[21]. The cuts impart new properties such as auxeticity[9,11], stretchability[8,10,15,22–24], conformability[8], multistability[25], and optical chirality[26], which have found broad applications in mechanical metamaterials[11,15,27,28], stretchable devices[8,10,23,29,30], 3D mechanical self-assembly[31], tunable adhesion[32], and soft machines[17,18,33].

Despite the advance, most studies focus on the local buckling of cut units in a thin sheet patterned with arrays of parallel slits or networked triangular or square cuts etc[8–13,15,17,18], generating quasi-3D pop-up structures without global curvatures. There are few studies on the shape shifting from a kirigami sheet to 3D shapes with intrinsic curvature[34–36]. Recent work shows that starting from a kirigami sheet or shell, 3D shapes with non-zero Gaussian curvature can be generated by utilizing either forward designs of non-periodic patterns of square cuts/cutouts[34,36] or inverse designs of tessellation of non-uniform square cuts patterning with irregular polygon cut units[35]. The local heterogeneous deformation among non-periodic tessellated cut units induces global out-of-plane buckling of the 2D kirigami sheets, thus, resulting in the formation of different 3D curved shapes[34,35]. However, it often requires programming intricate cut patterns and arrangements of non-periodic cut units, making the inverse design and optimization for target 3D shapes complicated and challenging[35,36]. In addition, how to utilize the 3D curved shapes in kirigami sheets for functionalities remains largely unexplored[34–36].

Theoretically, the curvature of a boundary can be harnessed to tune 3D curved shapes based on the classical Gauss-Bonnet theorem in differential geometry[37], which correlates the Gaussian curvature and the geodesic curvature along the boundary (i.e., the projection of boundary curvature). Motivated by this theorem, here, we propose a simple strategy of utilizing the boundary curvature of cut edges rather than complex cut patterns to program 3D curved shapes through both forward and inverse designs. In contrast to previous networked polygon cut units with square cut patterning[8,11,34–36], our kirigami sheet is composed of parallel discrete ribbons enclosed by continuous boundaries (Fig. 1a–c) through simple patterning of parallel cuts. We demonstrated that simply stretching the kirigami sheet with prescribed curved cut boundaries could generate varieties of well-predicted 3D curved shapes with positive, negative, and zero Gaussian curvatures and their combinations. We proposed a straightforward inverse design strategy for target 3D curved shapes, avoiding the necessity of shape optimization by building on top of theoretical insights from applying the Gauss–Bonnet theorem to the geodesic ribbons. Leveraging this, we demonstrated their potential applications in designing a universal gripper with dynamically programmable morphology for delicate objects and a biomimetic conformable heating pad with intrinsic adaptivity for human knees.

## Results

**Manipulating 2D boundary curvatures for 3D curved morphologies.** The classical Gauss–Bonnet theorem[37] correlates the boundary curvature with the global Gaussian curvature $K$. Motivated by the theorem, as shown in Fig. 1a–f, we start by designing the 2D precursors of kirigami sheets with different boundary curvatures $k_{bo}$ to exploit its effects on the Gaussian curvature of their 3D deployed shapes, where $k_{bo}$ is set to be positive (circular boundary in Fig. 1a), zero (rectangular boundary in Fig. 1b), and negative (biconcave circular boundary in Fig. 1c), respectively. We use the polyethylene terephthalate (PET) sheet with Young's modulus of 3.5 GPa, Poisson's ratio of 0.38, and thickness of 127 μm to fabricate the kirigami sheets using laser cutting (see "Methods" section). The thin sheets are cut into a number of discrete parallel thin ribbons enclosed by continuous boundary ribbon.

Figure 1d–f show that stretching the 2D precursors leads to distinct spheroidal, cylindrical, and saddle shapes with positive, zero, and negative Gaussian curvature $K$, respectively (Supplementary Movie 1). Upon stretching, the boundary ribbon starts bending and compresses the enclosed discrete ribbons to induce their out-of-plane buckling. Thus, it renders a 3D pop-up morphology. Once the 3D shape is formed, the global shape will not change but with its magnitude of curvature increasing with the applied strain. The three samples exhibit similar J-shaped force–displacement curves as shown in Fig. 1g, where the force increases approximately linearly with the initial displacement due to the bending-dominated deformation in the discrete ribbons, followed by the steep rise arising from the stretching-dominated deformation in the boundary ribbon. Such stiffness strengthening mechanical responses are similar to that observed in the kirigami sheet patterned with orthogonal square cuts[24]. Among the three samples, the circular one morphing into a spheroidal shape shows the highest stiffness and the least stretchability, while the biconcave one deforming into a saddle shape is the most compliant and stretchable (Fig. 1g).

We note that distinct from the kirigami sheets composed of networked polygon cut units in previous studies[8,11,34–36] or discrete structures composed of disconnected non-geodesic ribbons[38], the simple design of parallel cuts in this work endows the unique characteristic, i.e., parallel cuts make each discrete ribbon a geodesic curve of the morphed morphologies (see "Methods" section). It will facilitate the inverse design and dynamically programming morphologies, as discussed later.

By extending the classical Gauss–Bonnet theorem in differential geometry to the two neighboring enclosed ribbons and the multiple-connected enclosed kirigami surface morphology (Supplementary Fig. 1 and see "Methods" section), we can qualitatively explain the observed 3D curved shapes and their dynamic shape morphing. Mathematically, for the morphed 3D pop-up morphologies, the theorem can be simplified as

$$\int_{\Omega} K \mathrm{d}A + \oint_{\partial\Omega} k_{gb}\mathrm{d}s = C, \tag{1}$$

where the constant $C = 2\pi\chi(\Omega) - \sum_{i=1}^{p}\theta_i$ with $\chi(\Omega)$ and $\theta_i$ denoting the Euler characteristic of the Riemannian manifold $\Omega$ with boundary $\partial\Omega$ and the exterior angles at the vertices of the manifold, respectively. $C$ remains unchanged during shape shifting (see "Methods" section). $k_{gb} = k_b \sin\varphi$ is the geodesic curvature along the boundary ribbon according to the Meusnier theorem[37], i.e., the projection of the deformed boundary curvature $k_b$ with $\varphi$ being the projection angle (see "Methods" section).

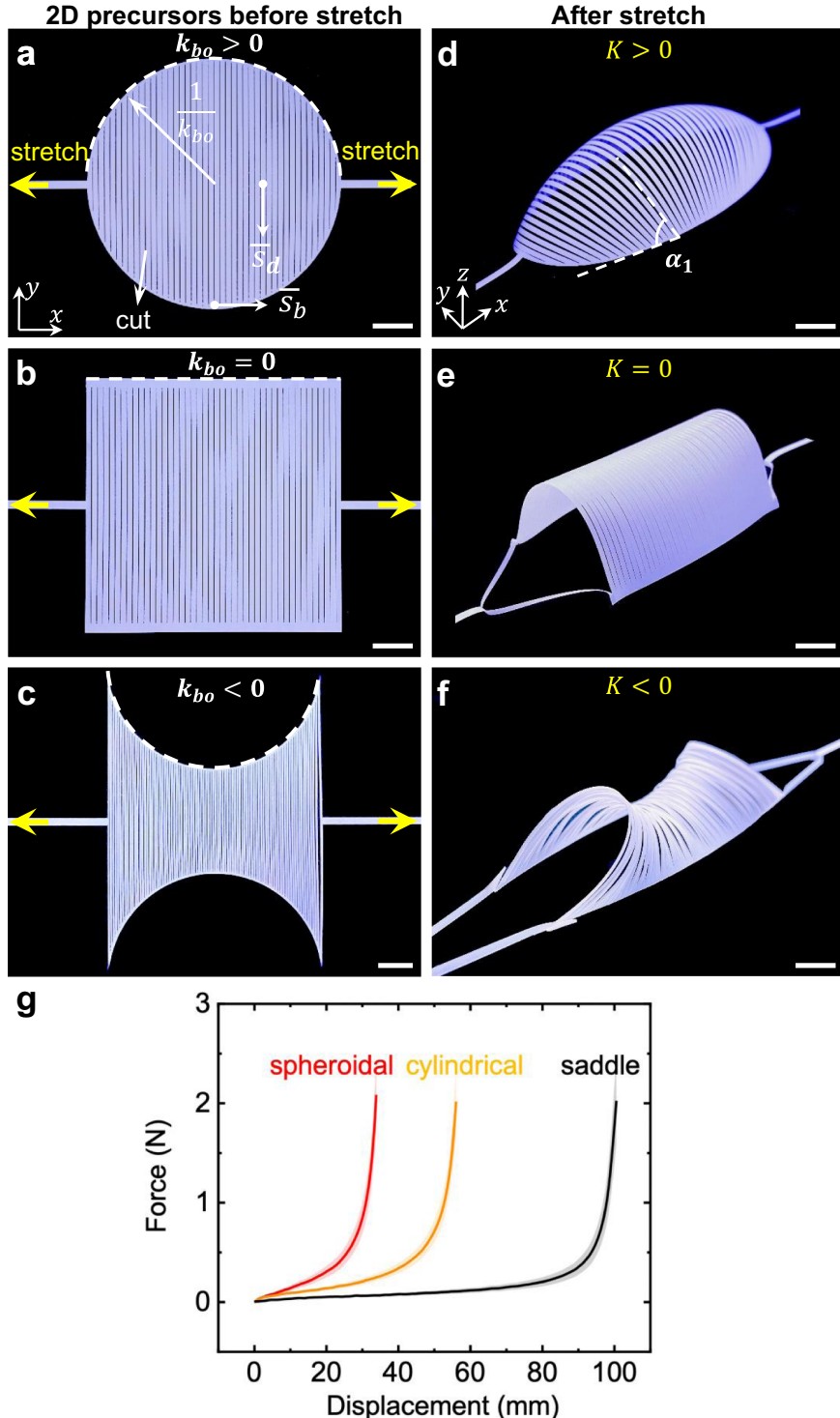

**Fig. 1 Shape shifting of 3D curved morphologies from 2D kirigami sheets with different cut boundary curvatures subject to uniaxial tension. a–c** 2D precursors of three kirigami sheets patterned with parallel cuts but different boundary curvatures $k_b$ highlighted in dashed white curves. circular (**a**), square (**b**), biconcave (**c**) samples with positive, zero, and negative boundary curvature, respectively. **d–f** The corresponding formed 3D curved shapes with different Gaussian curvature $K$. **d** Spheroidal shape with $K > 0$ at an applied strain of 0.30. **e** Cylindrical shape with $K = 0$ at an applied strain of 0.65. **f** Saddle shape with $K < 0$ at an applied strain of 1.47. Scale bars = 10 mm. **g** Force–displacement curves for the three 2D precursors. The shaded areas are the standard deviation between four different tests.

For the 2D kirigami precursor with positive boundary curvature, i.e., $k_{bo} > 0$, we have $C = \oint_{\partial\Omega_o} k_{gb}\,\mathrm{d}s$ by setting $K = 0$ with $\Omega_o$ denoting the manifold before deformation. After deformation, for the deformed manifold $\Omega'$, we have $\int_{\Omega'} K\,\mathrm{d}A = C - \oint_{\partial\Omega'} k_{gb}\,\mathrm{d}s = \oint_{\partial\Omega_o} k_{gb}\,\mathrm{d}s - \oint_{\partial\Omega'} k_{gb}\,\mathrm{d}s$ in terms of Eq. (1). As the applied strain increases, both $k_b$ and $\sin\varphi$ decrease, which results in a decreased geodesic curvature $k_{gb}$, and consequently $\int_{\Omega'} K\,\mathrm{d}A > 0$. Given the $C^2$ continuous

boundary curves in the three characteristic precursors ($C^2$ continuity means that both the first and second derivatives of the curves are continuous, i.e., continuous in curvature), both $\int_{\Omega'} K dA$ and $K$ will be simultaneously positive or negative. Thus, we have a globally positive $K$ in the deformed manifold $\Omega'$, i.e., $K > 0$ in $\Omega'$, which is consistent with the observed spheroidal shape in Fig. 1d. Similarly, for the 2D precursor with $k_{bo} < 0$, as the strain increases, the absolute value of the boundary curvature $|k_b|$ becomes smaller and $\sin \varphi$ decreases, which results in an increased geodesic curvature. Thus, we have the generated saddle shape with globally $K < 0$ in Fig. 1f. For the 2D precursor with $k_{bo} = 0$, during the deformation, $k_{bo} = 0$ does not change, which leads to a zero geodesic curvature. Thus, we have a cylindrical shape with $K = 0$ in Fig. 1e.

**Analytical modeling and simulation on 3D shape shifting.** To quantify the shape shifting of the kirigami structures with the applied strain, we combine both analytical modeling and finite element method (FEM) simulation to predict their morphology changes (see "Methods" section). The deformation of the kirigami structures is dominated by bending of the discrete ribbons, where the elastic strain energy in the boundary ribbon is negligible due to its small width (Supplementary note 1). Thus, all the discrete ribbons share similar deformed elastica shapes[39–42]. The deformed 3D shape at an applied strain $\varepsilon$ can be described by $\mathbf{r}_s(\bar{s}_b, \bar{s}_d) = (\bar{x}(\bar{s}_b, \bar{s}_d), \bar{y}(\bar{s}_b, \bar{s}_d), \bar{z}(\bar{s}_b, \bar{s}_d))$, where $\bar{s}_b$ and $\bar{s}_d$ denote the normalized arc length coordinate of the boundary and the discrete ribbon as illustrated in Fig. 1a, respectively. $(\bar{x}, \bar{y}, \bar{z})$ denote the Cartesian coordinates of any point $P(\bar{s}_b, \bar{s}_d)$ on the surface with its origin set at the center of the 2D precursor. Considering the deformed surface shape foliated by continuously varying discrete ribbons along the boundary, its generalized shape functions can be expressed as (see details in Supplementary note 1)

$$\bar{x}(\bar{s}_b, \bar{s}_d) = \frac{2m}{\lambda} CN(\lambda \bar{s}_d, m) \cos \alpha_1 + f(\bar{s}_b, \varepsilon), \quad (2)$$

$$\bar{y}(\bar{s}_b, \bar{s}_d) = \frac{2}{\lambda} E(AM(\lambda \bar{s}_d, m), m) - \bar{s}_d, \quad (3)$$

$$\bar{z}(\bar{s}_b, \bar{s}_d) = \frac{2m}{\lambda} CN(\lambda \bar{s}_d, m) \sin \alpha_1, \quad (4)$$

by sweeping the varying discrete ribbons modeled as an elastica shape along the boundary. $m = m(\bar{s}_b, \varepsilon)$ is the elliptical modulus that characterizes the bending deformation of a discrete ribbon. $\lambda = 2F(\frac{\pi}{2}, m)/\bar{l}_d$ is related to the normalized length $\bar{l}_d$ of the discrete ribbon. $AM$ and $CN$ denote the Jacobian amplitude and the elliptic cosine, respectively. $E$ and $F$ denote the incomplete elliptic integral of the second kind and the first kind, respectively. $\alpha_1 = \alpha_1(\bar{s}_b, \varepsilon)$ is the tilting angle of the discrete ribbon with respect to the horizontal plane (i.e., $xy$ plane) as shown in Fig. 1d, which varies from 0 to 180° depending on its boundary location and the applied strain. $f(\bar{s}_b, \varepsilon)$ describes the $x$ coordinate at $\bar{s}_b$ of the deformed boundary ribbon.

Without losing generality, we can use three profiles from the front view, top view, and side view to characterize the 3D shape shifting with the applied strain (Fig. 2a–c for spheroidal shapes, Fig. 2e–g for saddle shapes and Supplementary Fig. 5a–c for cylindrical shapes). The front view shows the backbone profile on the $xz$ plane (Fig. 2a, e and Supplementary Fig. 5a), which can be predicted by $\bar{x}_{bb} = \frac{2m}{\lambda} \cos \alpha_1 + f(\bar{s}_b, \varepsilon)$ and $\bar{z}_{bb} = \frac{2m}{\lambda} \sin \alpha_1$ after setting $\bar{s}_d = 0$ and $\bar{y} = 0$ in Eqs. (2–4). The top-view profile shows the deformed shape of the boundary ribbon (Fig. 2b, f and Supplementary Fig. 5b) that remains in

the $xy$ plane during deformation by setting $\bar{z} = 0$ in Eqs. (2–4), which can be parametrized by

$$\mathbf{r}_b(\bar{s}_b, \varepsilon) = (\bar{x}, \bar{y}, 0) = (f(\bar{s}_b, \varepsilon), g(\bar{s}_b, \varepsilon), 0), \quad (5)$$

where

$$g(\bar{s}_b, \varepsilon) = \left[ \frac{2E(\frac{\pi}{2}, m)}{F(\frac{\pi}{2}, m)} - 1 \right] g(\bar{s}_b, 0), \quad (6)$$

describes the $y$ coordinate at $\bar{s}_b$ of the deformed boundary ribbon at the strain of $\varepsilon$. Equation (6) describes the relationship between $m$ and $\varepsilon$. Thus, combining Eqs. (2–4) and Eq. (6) will determine the unknown parameters of $\bar{x}, \bar{y}, \bar{z}$, and $m$ to predict the deformed 3D shapes with the applied strain. The side view shows the projection of similar elastica shapes of discrete ribbons onto the $yz$ plane (Fig. 2c, g and Supplementary Fig. 5c), which depends on $m$ and the tilting angle of the longest discrete ribbon. Its deformed elastica shape can be expressed by $\bar{y}_d = \frac{2}{\lambda} E(AM(\lambda \bar{s}_d, m), m) - \bar{s}_d$ and $\bar{z}_d = \frac{2m}{\lambda} CN(\lambda \bar{s}_d, m)$, where the length of the discrete ribbons is assumed to be unchanged during deformation.

Next, we apply both the generalized analytical model and FEM simulation to analyze the 3D shape shifting in the specific examples shown in Fig. 1. Figure 2a–c theoretically predict the variation of the three profiled shapes with the applied strain $\varepsilon$ during the formation of a spheroidal shape. As $\varepsilon$ increases from 0 to 0.4, top-view profiles show that the circular boundary gradually deforms into an elliptical shape (Fig. 2b), where we have $f(\bar{s}_b, \varepsilon) = (1 - \bar{w}) \sin \bar{s}_b + \bar{v} \cos \bar{s}_b$ and $g(\bar{s}_b, \varepsilon) = (1 - \bar{w}) \cos \bar{s}_b - \bar{v} \sin \bar{s}_b$ ($\bar{s}_b \in [-\frac{\pi}{2}, \frac{\pi}{2}]$) in the model (Supplementary note 1). $\bar{w}(\bar{s}_b, \varepsilon)$ and $\bar{v}(\bar{s}_b, \varepsilon)$ denote the radial and tangential displacement of the boundary ribbon[43], respectively. Correspondingly, the compressed discrete ribbons deform into an elastica shape (side view in Fig. 2c). The backbone expands and shows an elliptical profile (front view in Fig. 2a and Supplementary note 1). As shown in Fig. 2a–c, the superposition of the three theoretically predicted front-view, top-view, and side-view profiles (highlighted in purple color) with images retrieved from the experimental observation at $\varepsilon = 0.3$ shows an excellent agreement. The corresponding FEM simulated deformed 3D shape shows an excellent overlapping with the experiment (Fig. 2d). Differently, during the formation of the cylindrical shape, both the boundary ribbon and backbone profile remain straight during deformation and all the discrete ribbons take the same elastic shape, the modeling of which is consistent with both experiments and FEM simulation (Supplementary Fig. 5).

Figure 2e–g show the predicted shape change during the formation of a saddle shape. In contrast to simultaneous buckling in generating the spheroidal and cylindrical shapes, we observe a sequential buckling during the formation of the saddle shape in experiments (Supplementary Movie 1). The discrete ribbons near two stretching ends pop up first, followed by the ribbons in the center when beyond a critical strain $\varepsilon_c$ (Supplementary Fig. 6). The physical origin of the sequential buckling is due to the coupling effects of the concave boundary geometry and different critical buckling forces of the discrete ribbons (Supplementary note 1), where the curvature varies sequentially during deformation along the boundary ribbon from its two ends to the center (Fig. 2f). Such sequential buckling behavior disappears for the large radius of curvature since the 2D precursor is close to a rectangle shape. As the applied strain further increases, the discrete ribbons contact with each other, leading to structural frustration. Reducing the number of ribbons facilitates a frustration-free structure without self-contact (Supplementary Fig. 7). Such a sequential shape shifting is well captured by both the analytical model and FEM simulation. As predicted by the

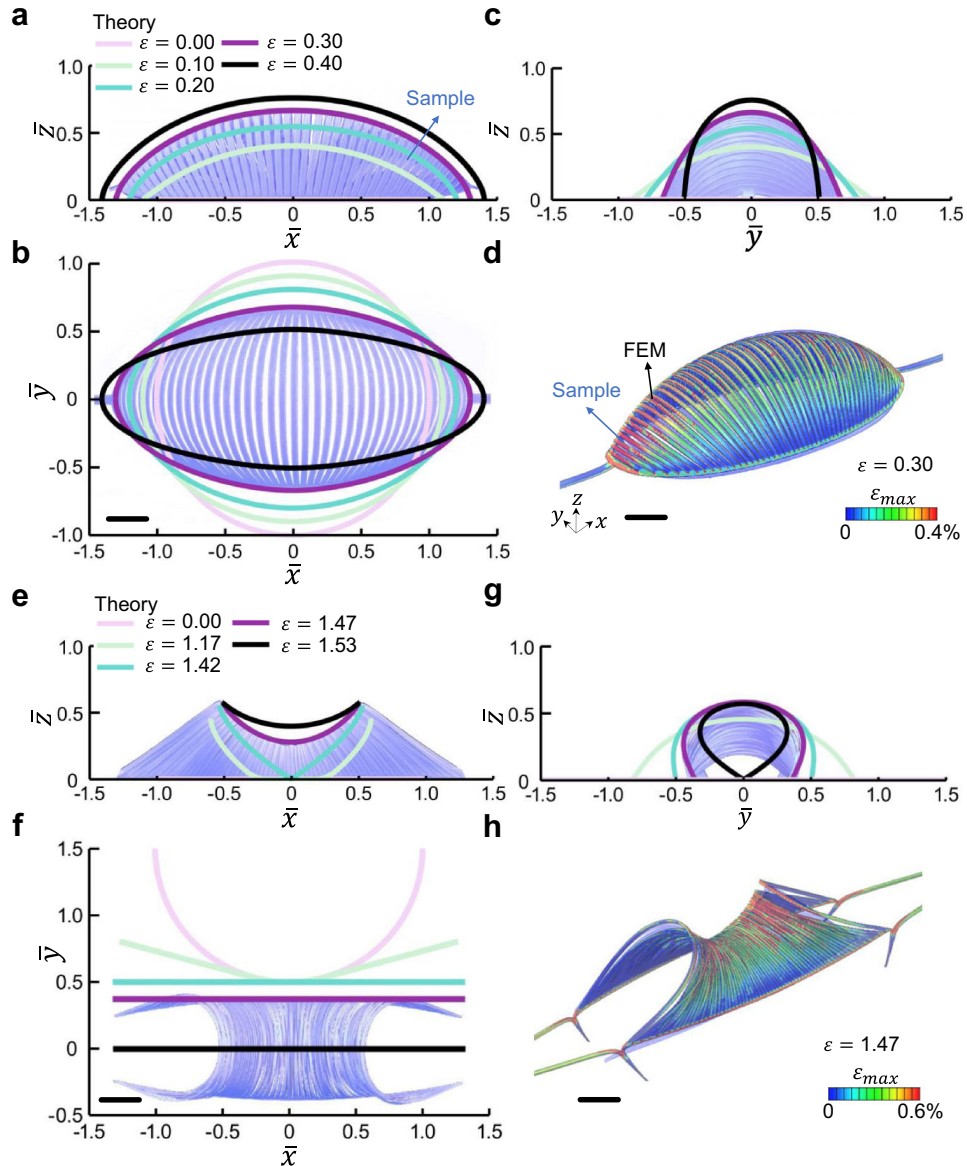

**Fig. 2 Quantifying the 3D shape shifting through analytical modeling and simulation. a–h** Predicted shape changes with the applied strain $\varepsilon$ in the samples of spheroidal (**a–d**) and saddle shapes (**e–h**). **a, e** Front-view profile. **b, f** Top-view profile. **c, g** Side-view profile. **d, h** Overlapping of FEM simulation results (contours of the maximum principal strain $\varepsilon_{max}$) with the experimental image at $\varepsilon = 0.30$ (**d**) and 1.47 (**h**). Scale bars = 10 mm.

model, Fig. 2f shows that at the critical strain $\varepsilon_c \approx 1.42$, the initial concave boundary ribbon deforms into a straight line and remains straight upon further deformation, where we have $f(\bar{s}_b, \varepsilon) = \bar{s}_b$ and $g(\bar{s}_b, \varepsilon) = \sqrt{1.46 - (\varepsilon - 0.32)^2}$ with $\bar{s}_b \in [-1.32, 1.32]$ in the model (Supplementary Note 1). Correspondingly, the backbone profile (Fig. 2e) transits from an initial sharp V shape to a smooth concave shape, which exhibits a sudden jump of the displacement along the $z$-axis when the applied strain is slightly beyond $\varepsilon_c$. Further stretching results in the formation of the saddle shape with a concave backbone, which is consistent with both experiments (Fig. 2e–g) and FEM simulation results (Fig. 2h).

**Quantitative correlation between the boundary curvature and the Gaussian curvature.** Based on the validated theoretical model, we further establish the general quantitative correlation between the boundary curvature $k_{bo}$ of 2D kirigami precursors

and the Gaussian curvature $K$ of their popped 3D morphologies at a given applied strain (see details in Supplementary Note 2). Figure 3a, b show the theoretically predicted 3D maps of the normalized Gaussian curvature $\bar{K}$ at the center point $C(\bar{s}_b = 0, \bar{s}_d = 0)$ as a function of both normalized boundary curvature $\bar{k}_{bo}$ (see illustration of tuning different $k_{bo}$ in the insets of Fig. 3a, b) and applied strain $\bar{\varepsilon}$. It shows that for 2D kirigami precursors with either positive (Fig. 3a) or negative boundary curvature (Fig. 3b), generally, the absolute value of $\bar{K}$ increases with an increasing strain $\bar{\varepsilon}$ and $|\bar{k}_{bo}|$. Note that for the formed saddle shapes, we have $\bar{K} = 0$ before reaching the critical strain $\varepsilon_c$. At the onset of $\varepsilon_c$, $\bar{K}$ suddenly decreases due to a dramatic increase in the boundary curvature. Beyond $\varepsilon_c$, $\bar{K}$ barely changes because the boundary ribbon remains straight (Fig. 3b). Interestingly, Fig. 3c shows that theoretically, the normalized variation of Gaussian curvature $|\Delta\bar{K}/\bar{K}_{max}|$ increases approximately linearly with the normalized variation

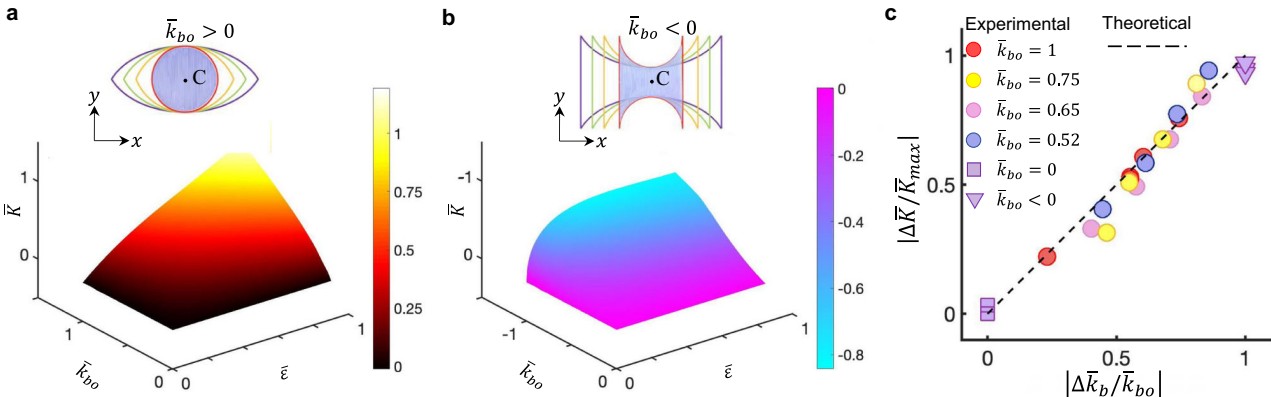

**Fig. 3 Quantifying the correlation between the boundary curvature and the Gaussian curvature. a, b** Theoretically predicted 3D maps of the normalized Gaussian curvature $\bar{K}$ at the center point $C$ as a function of the normalized boundary curvature $\bar{k}_{bo}$ in 2D kirigami precursors (insets) and the applied strain $\varepsilon$ for the cases of spherical (**a**) and saddle (**b**) shapes. The color bars represent the normalized Gaussian curvature. **c** Theoretical and experimental results of the approximately linear relationship between the normalized variation of the Gaussian curvature $|\Delta\bar{K}/\bar{K}_{max}|$ and the normalized variation of the boundary curvature $|\Delta\bar{k}_b/\bar{k}_{bo}|$.

of boundary curvature $|\Delta\bar{k}_b/\bar{k}_{bo}|$(slope $\approx$ 1) (Supplementary note 2), i.e.,

$$\frac{|\Delta\bar{K}/\bar{K}_{max}|}{|\Delta\bar{k}_b/\bar{k}_{bo}|} = \frac{4m^2\bar{k}_{bo}}{[\bar{k}_{bo}\bar{x}^2(\gamma/\bar{k}_{bo},0) - \bar{y}(0,1)]\bar{K}_{max}} \approx 1, \quad (7)$$

which is consistent with the experimental measurements. $\bar{K}_{max}$ is the maximum Gaussian curvature at the center point, and $\gamma$ denotes half of the central angle of the boundary curve. Note that there are few data points for the square and biconcave shapes because the cylinder and the saddle shape have zero and sudden-jumping Gaussian curvature, respectively. Specifically, this near-linear relationship holds regardless of the initial boundary curvature of a 2D kirigami precursor, which can be harnessed to program the morphology and the dynamic deployment trajectories.

**Combinatorial designs for more complex 3D shapes.** Next, equipped with the knowledge of the correlation between the boundary curvature and the deformed 3D shapes, we extend it to achieve more varieties of 3D shape-morphing morphologies through tuning the smoothness of the boundary curves of individual units in combinatorial and tessellated designs (Fig. 4a–j).

We first explore the 3D shape shifting in 2D kirigami precursors through tessellating the three basic units in Fig. 1a±c. Figure 4a shows a 2D diamond-shaped kirigami precursor composed of tessellated 2 × 2 square units with zero boundary curvature. Each unit has the same parallel cut pattern. Upon vertically stretching the 2D diamond precursor along the *y*-axis, both top and bottom square units pop up spontaneously (represented by the symbol of "+" in the inset of Fig. 4b) via out-of-plane buckling while the square units on two sides pop down (denoted by the symbol of "−"), generating a smiley 3D human face-like morphology (Fig. 4b). Note that both the inner and outer boundaries of an individual unit in the combinatorial design still belong to the cut boundaries following our model, both of which contribute to the geodesic curvature $k_{gb}$ in Eq. (1) and satisfy the Gauss–Bonnet theorem locally and globally. Specifically, the eyes and mouth in the form of a hole are formed due to its discontinuous slope (changed smoothness) at the intersections of boundaries, which results in a localized non-zero

Gaussian curvature. This is also consistent with the Gauss-Bonnet theorem, where the localized non-zero Gaussian curvature arises from the localized curvature change (variation of the exterior angle in Eq. (1)) of the $C^0$ smooth inner and outer boundary curves of the units (Supplementary Note 3). The holes divide the face into eight independent popping regions (e.g., forehead, eyes, nose, cheek, mouth, and chin). We note that the face will not change its pattern under different loading rates. Similarly, stretching an array of 3 × 1 rectangle units with identical parallel cuts and zero boundary curvature (Fig. 4e) along the *x*-axis leads to a sinusoidal wavy shape (Fig. 4f) with zero Gaussian curvature. Furthermore, Fig. 4h shows that two circular units with positive boundary curvatures bridged with a biconcave unit with negative boundary curvatures form a vertically tessellated 2D precursor. Stretching the 2D precursor along the *x*-axis generates an increasingly enclosing 3D shape, where the two circular units pop up into a spheroidal shape with positive Gaussian curvature while the concave unit buckle into a saddle shape with negative Gaussian curvature (Fig. 4i). As the stretching strain further increases, their two end circular boundaries contact with each other, forming an encapsulated Venus flytrap-like structure (Fig. 4j) that could be used for delicate grippers, as discussed later.

We note that given the combinatorial design of units and the bistability in each unit, the formed 3D shapes in the kirigami sheets could be further reconfigured to other distinct 3D morphologies via the bistability switch[25] in the buckled discrete ribbons locally or globally, where each ribbon could pop up or pop down independently and locally as shown in Supplementary Fig. 10. For example, manipulating the bistable switch in eight independent popping regions of the human face-like morphology in Fig. 4b, i.e., the popping directions of discrete ribbons in each region, could generate more potential facial expressions. As a proof of concept, manually flipping all the popping directions in the eight regions of the smiley face under the stretched state generates a sad face (Fig. 4c), which can be reversibly switched to the smiley face. Furthermore, localized flipping of two single ribbons in the eye area generates a face with eyeglasses (Fig. 4d). The bistable states of the ribbons could be either manually switched[25] or potentially remotely tuned using the magnetic field (Supplementary Fig. 11, Supplementary Movie 2, and Supplementary Note 4). Similarly, Fig. 4g shows that flipping the popping direction in the central unit of the sinusoidal wavy shape

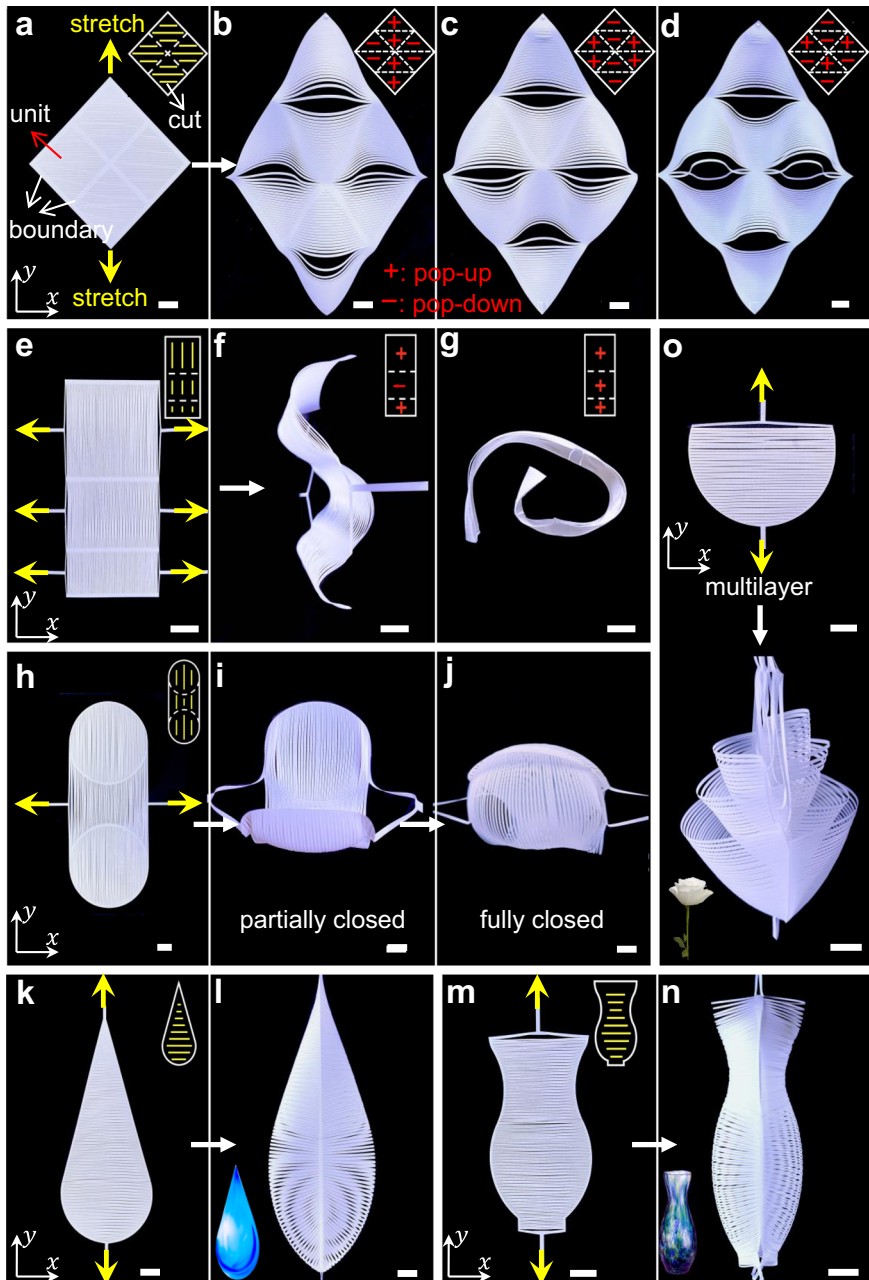

**Fig. 4 Combinatorial designs of 2D kirigami precursors for complex 3D shapes under uniaxial tension. a–j** Reconfigurable 3D shapes through bistability of discrete ribbons. **a** 2D precursor composed of 2 × 2 square units with zero boundary curvature. **b–d** Uni-axial stretching induced reconfigurable human face-like morphologies with switchable smiley (**b**) and sad (**c**) expressions, as well as eyeglasses (**d**) by tuning the popping directions of ribbons (insets). **e** 2D precursor composed of 3 × 1 rectangle units with zero boundary curvature. **f, g** Formation of switchable sinusoidal wavy and coiled shapes. **h** 2D precursor composed of an array of two circular units bridged with a biconcave unit. **i, j** Formation of a 3D encapsulated Venus flytrap-like shape through uniaxial stretching. **k–n** Formation of a 3D droplet-like shape (**l**) and vase-like shape (**n**) by uni-axially stretching 2D precursors with different combined boundary curvatures (**k, m**). Insets show the image of a droplet and a vase. **o** Formation of a flower-like shape by uni-axially stretching multiple layers of semi-circular 2D precursors. Scale bars = 10 mm.

in Fig. 4f makes it reconfigure into a coiled shape, which could be reversibly switched by flipping back the popping direction.

More complex 3D shapes can be generated by combining boundary curvatures with different smoothness in stacked 2D kirigami precursors (Fig. 4k–n) under uni-axial mechanical stretching, e.g., a 3D droplet-like shape (Fig. 4l) and a vase-like shape (Fig. 4n). We note that the smoothness of the backbone in the generated 3D shapes is controlled by the smoothness of the boundary in their corresponding 2D precursors, which

makes the combinatorial design easy to handle. To generate the water droplet shape, we design a 2D precursor consisting of combined a straight line and a circular arc to mimic the 3D water droplet's backbone shape (Fig. 4k). Similarly, the vase-like shape (Fig. 4n) is generated by stretching two stacked 2D precursors composed of a concave and convex boundary (Fig. 4m). Furthermore, stretching multiple layers of similar semi-circular 2D precursors generates a flower-like shape with multilayer pedals (Fig. 4o).

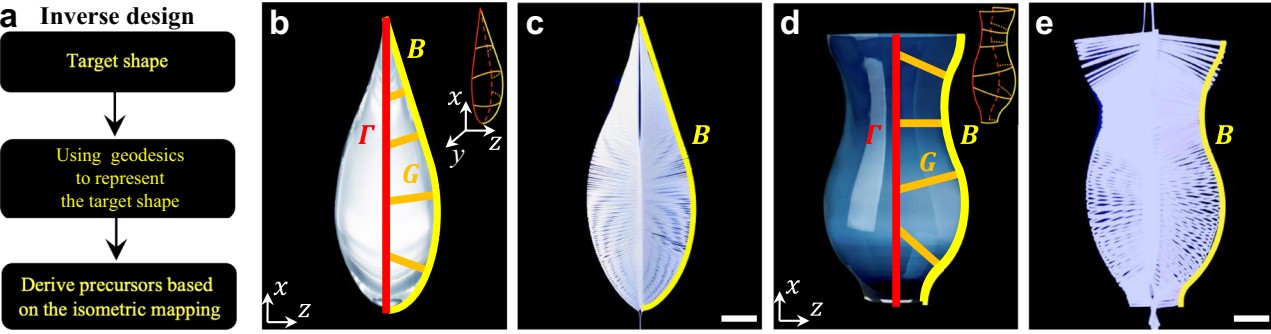

**Fig. 5 Inverse design of 3D shapes. a** Flow diagram of the inverse design. **b** Schematic of using curves to approximate and represent the target shape (side view of a waterdrop). The inset shows an isometric view. Red, orange, and yellow curves are the boundary curve $\Gamma$, geodesic curve $G$, and backbone curve $B$, respectively. **c** Experimental inverse-design result of the waterdrop shape formed by a 2D kirigami precursor subject to uniaxial tension. The yellow curve is the backbone in the target shape. **d** Schematic of using curves to approximate and represent the target shape (side view of a vase), with an isometric view showing in the inset. **e** Experimental inverse-design result of the vase shape formed by a 2D precursor subject to uniaxial tension. Scale bars = 10 mm.

**Inverse design strategy**. Existing methods of inverse design for target 3D-curved shapes using the kirigami approach require complex algorithms to program heterogeneous local deformation among networked cut units[35,36]. Based on the information that discrete ribbons are geodesic curves of the deformed 3D shapes, we propose a straightforward inverse design strategy. It utilizes the geodesic curves extracted from the target shapes and the isometric mapping to prescribe the 2D precursors (Fig. 5a), which is, in principle, applicable to any target configuration.

To illustrate the strategy, we use the target shapes of a water droplet (Fig. 5b) and a vase (Fig. 5d) as two examples for the inverse design of the 2D kirigami precursors. As shown in Fig. 5b, d, we first approximate and represent the target shapes by deriving the shape functions of the backbone curve $B$ (highlighted in yellow color, see details in Supplementary Note 5), the geodesic curves $G$ (highlighted in orange color) approximated by elastica curves, and the boundary curve $\Gamma$ (highlighted in red color). Next, based on the isometric mapping from $G$ and $\Gamma$ in the target shape to the 2D precursor, we derive the shape function of the prescribed 2D boundary curve $\Gamma^{\mathrm{P}}$ (Supplementary Note 5). The parametrization of $\Gamma$ and $\Gamma^{\mathrm{P}}$ can be expressed in the form of $\boldsymbol{r}_{\Gamma} = (x(s_{\mathrm{b}}), y(s_{\mathrm{b}}), 0)$ and $\boldsymbol{r}_{\Gamma^{\mathrm{P}}} = (x^{\mathrm{P}}(s_{\mathrm{b}}), y^{\mathrm{P}}(s_{\mathrm{b}}), 0)$, respectively, where the superscript $P$ represents the 2D precursor. Accordingly, the shape function of the boundary curve $\Gamma^{\mathrm{P}}$ in the 2D precursor can be derived as (see details in Supplementary Note 5)

$$\left\{ \begin{array}{c} x^{\mathrm{P}}(s_{\mathrm{b}}) \\ y^{\mathrm{P}}(s_{\mathrm{b}}) \end{array} \right\} = \begin{bmatrix} \eta_{\mathrm{x}} & 0 \\ 0 & \eta_{\mathrm{y}} \end{bmatrix} \left\{ \begin{array}{c} x(s_{\mathrm{b}}) \\ y(s_{\mathrm{b}}) \end{array} \right\}, \tag{8}$$

where the parameters $\eta_{\mathrm{x}}$ and $\eta_{\mathrm{y}}$ are related to the isometric mapping. Furthermore, the required strain $\varepsilon_{re}$ to form the target shape is given by $\varepsilon_{re} = [x(\max(s_{\mathrm{b}})) - x^{\mathrm{P}}(\max(s_{\mathrm{b}}))]/x^{\mathrm{P}}(\max(s_{\mathrm{b}}))$ with $\max(s_{\mathrm{b}})$ being the maximum arc length of the boundary ribbon.

Figure 5c, e show the result of the inverse design of a water droplet and a vase after deploying the derived 2D kirigami precursors at an applied strain of $\varepsilon_{re} = 0.14$ and 0.07, respectively. The inverse design result agrees well with the target shape denoted by the yellow curves. We note that for a target discrete 3D-curved shape, previous studies using the assembly of disconnected non-geodesic ribbons without an enclosed boundary need complicated control and optimization of geometry and distribution of each ribbon locally and globally in the inverse design[38]. Notably, precise control of all the geodesic ribbons and shape optimization are not necessary for our proposed inverse design approach, since it harnesses the isometric mapping of geodesics and only needs the information

of one representative geodesic curve and one boundary curve in the target surface. Thus, such a strategy could significantly simplify the calculation (Supplementary Note 5). Moreover, programming the dynamic shape-shifting morphology of the global geometric entity in our design is limited to only one variable, i.e., the boundary curvature in terms of Eq. (7). Only geodesic ribbons have zero geodesic curvatures, and their curvatures are the normal curvatures on the morphed surface, where the easy-to-handle global dynamic program arises naturally. It does not require the trivial control and optimization of the heterogeneous deformation of the polygons in networked kirigami structures[35,36] or the thickness distribution of each non-geodesic ribbon in a discretized manner[38].

Next, based on the revealed design principle on utilizing boundary curvature for programmable shape morphing in the kirigami sheets, we further explore the potential of harnessing dynamic shape morphing for shape-determined multifunctionality. To show their potential, we demonstrate two proof-of-concept examples with their potential applications in delicate flexible grippers and conformable heaters.

**Application: delicate and noninvasive kirigami hand designed via the programmable morphology**. A simple, rapid, and economical soft gripper is highly required in biomedical robotics and wildlife-conservation devices. However, for the existing soft grippers realized by pneumatic[44–46], hydraulic[47], and magnetic actuation[48], and responsive materials[17] using pinching[44,45,48], enclosing[47], and suction[49], it is challenging to balance the response time, manufacturing cost, simplicity of designs, and robustness in noninvasive grasping missions. Here, utilizing the dynamically programmable shape morphing, we present a universal, flexible yet robust kirigami hand, which can encapsulate gelatinous and delicate organisms nondestructively in unstructured environments.

Subject to simply uniaxial stretch, the 2D kirigami precursor composed of two circular units bridged with a biconcave unit (Figs. 4h and 6a) transforms into an encapsulated Venus-flytrap-like shape composed of two hemispheres bridged with a saddle shape (Fig. 6b). To demonstrate its delicacy in noninvasively grasping extremely soft and slippery objects, we use the example of grasping a raw egg yolk from a petri dish with the grasping process and mechanism from open to closed states shown in Fig. 6c–e in both side view and front view (insets) and Supplementary Movie 3. First, the biconcave unit starts bending and forms a V shape with the increasing uniaxial stretch (Fig. 6c). Next, as shown in Fig. 6d, the two circular units transform into two hemispheres with the variation of the boundary curvature

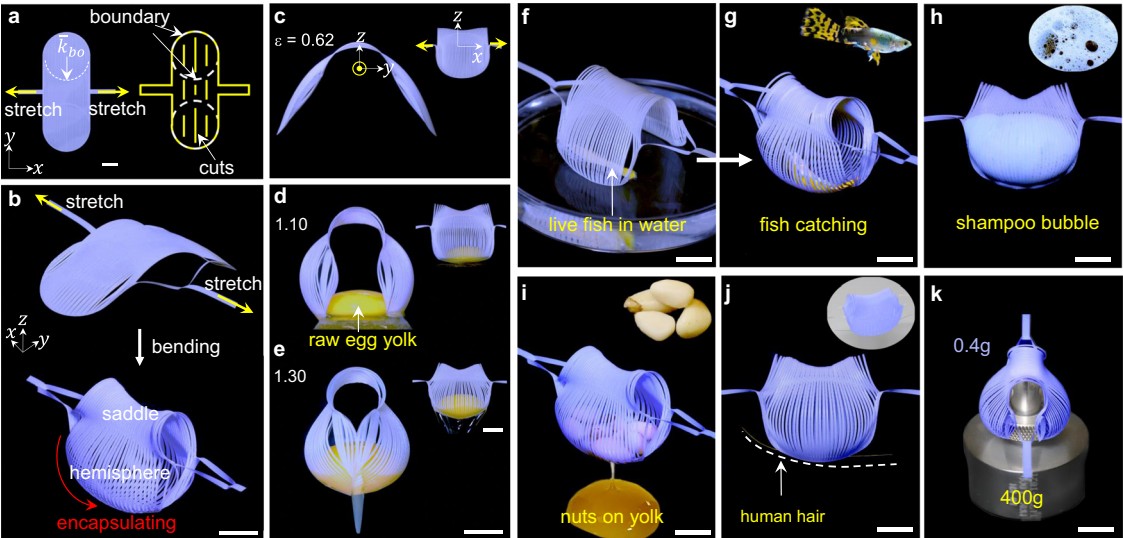

**Fig. 6 Programmable delicate and noninvasive kirigami gripper. a** 2D precursor composed of an array of two circular units bridged with a biconcave unit. $\bar{k}_{bo}$ defines the initial boundary curvature of the units. Yellow arrows are the direction of the uniaxial stretching. Yellow lines and white dashed lines represent the cuts and the boundaries, respectively. **b** Isometric view of the morphology from bending to encapsulating upon stretching. Red arrows represent the morphing direction of the hemisphere. **c**–**e** Side views of the grasping process of a raw quail egg yolk with the increasing applied strain from 0.62 to 1.3, respectively. The inset shows the corresponding front views. **f, g** Encapsulating a live fish from a petri dish filled with water. **h** Grasping the super-soft shampoo bubbles from the surface of the water. **i** Collecting the granular objects (pine nuts) from the super-soft surface of a raw egg yolk. **j, k** Grasping a human hair (**j**) and a deadweight (400 g, **k**). Scale bars = 10 mm.

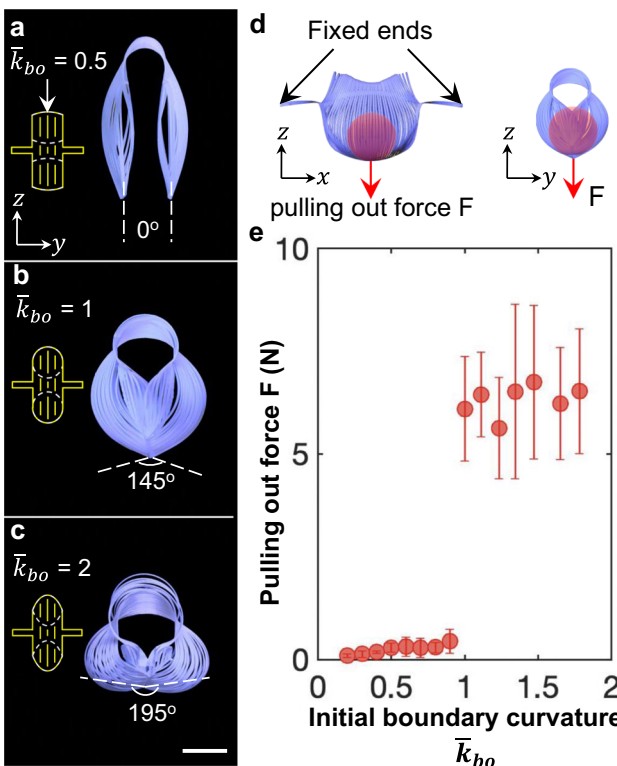

**Fig. 7 Effect of the initial boundary curvature on the pulling-out force of kirigami grippers. a**–**c** Experimental illustration of shapes formed by 2D precursors with the different initial-boundary curvatures $\bar{k}_{bo}$ at the maximum applied strain. The insets show the schematic figure of the 2D precursors. The white dashed line represents the angle between the tips of the gripper. **d** Schematic illustration of measuring the pulling-out force $F$ via pulling out a red sphere from the grippers with various boundary curvatures $\bar{k}_{bo}$. Red arrows are the direction of the pulling-out force. **e** The experimental results on the curve of $F$ vs. $\bar{k}_{bo}$. The error bars represent the standard errors of the mean.

and start grasping the egg yolk from the bottom. Last, the flattened boundary curve leads to the closure of the structure and encapsulating the egg yolk (Fig. 6e), where the super-slippery and soft yolk can be held for hours showing both the soft yet robust ability to encapsulate and preserve gelatinous organisms.

Further, to demonstrate the advantage of the bio-interactive hand, we rapidly encapsulate a live fish from water before it could escape (Fig. 6f, g and Supplementary Movie 3); we then release it unharmed. The noninvasive interaction shows the conformability and adaptivity of the gripper. Also, encapsulating super-soft objects (e.g., shampoo bubbles in Fig. 6h) and the collection of granular objects (e.g., pine nuts in Fig. 6i) from a super-soft substrate (e.g., a raw egg yolk) are demonstrated, broadening its versatility and noninvasiveness. Moreover, the universal gripper can be applied to a wide range of targets, including small objects such as a human hair (Fig. 6j), a coin, a thin micro-SD card, and blueberries, etc. (Supplementary Movie 3), as well as a 400 g deadweight (Fig. 6k) that is 1000 times the weight of the gripper (0.4 g). We note that despite the simple design, the gripper kirigami is capable of repeatedly lifting the 400 g deadweight for over 1400 cycles without causing materials and structural failure and sacrificing its grasping performance, demonstrating the robustness of the gripper.

The dynamic morphology programmed via tuning the boundary curvature can be further harnessed to tailor the holding force of the flexible gripper[47]. As shown in Fig. 7a–e, when the normalized initial-boundary curvature $\bar{k}_{bo}$ (illustrated in Fig. 7a) of the 2D precursors increases from 0 to 2, after simple stretching, the angle formed by the two tips of the hemispheres increases from 0 to 195°, correspondingly, their final deformed shapes transit from open ($\bar{k}_{bo} < 1$ in Fig. 7a) to closed ($\bar{k}_{bo} \geq 1$ in Fig. 7b, c). The precursor with $\bar{k}_{bo} = 1$ defines a critical state, where the two curved ends can become contacted to form a 3D encapsulated shape (Fig. 7b). Correspondingly, as shown in Fig. 7d, e on the pulling force of the kirigami grippers vs. $\bar{k}_{bo}$, it results in a sudden jump of the pulling-out force of the kirigami grippers at $\bar{k}_{bo} = 1$ from an average force of 0.4 N ($\bar{k}_{bo} < 1$) to

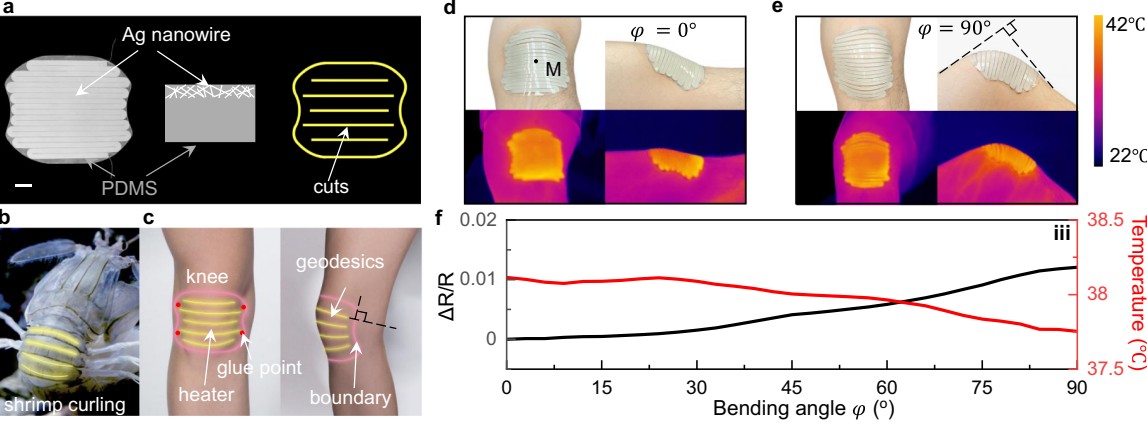

**Fig. 8 Dynamically conformable biomimetic heater. a** Photograph of the 2D precursor. Scale bar: 10 mm. The right shows the schematics of the cross-section of the AgNW/PDMS heater and the cut pattern. **b**, **c** Schematics of the heater mimicking the shell of the Mantis shrimp composed of geodesics attached to the human knee. **b** Curling shell of the Mantis shrimp. Yellow lines represent geodesics. **c** Schematic of the heater attached to the knee. Pink and yellow lines are the boundary and the geodesics, respectively. Red dots are the points with the adhesive. Black dashed lines representing the geodesics are perpendicular to the boundary with the knee bending. **d**, **e** The kirigami heater deforms with the knee as the knee bends from 0° (**d**) to 90° (**e**) and the corresponding thermal image upon heating. $\varphi$ denotes the bending angle of the knee. M is the center of the knee. The color bar represents the temperature. **f** Electric resistance of the heater and the temperature of the point M as a function of the bending angle $\varphi$.

6.1 N ($\bar{k}_{bo} \geq 1$), a 15 times grasping force enhancement. The pulling force is defined as the minimum force required to pull the red sphere out of the gripper, as schematically illustrated in Fig. 7d. The force jump is because the grasping mode transits from pinching by friction to distinct encapsulating due to the programmed shape of the grippers by the boundary curvature.

Further increasing the normalized boundary curvature beyond 2 does not lead to a higher pulling force (Supplementary Fig. 12b), since all the grippers share a similar closed shape under the same encapsulating grasping mode (Supplementary Fig. 12a), where further deformation is constrained by the contacted hemisphere petals. Furthermore, for the kirigami grippers with the same size and geometry, when reducing the number of parallel cuts or equivalently increasing the ribbon width (Supplementary Fig. 13a–e), we observe the similar sudden jumping of the dramatically reduced pulling force at a critical ribbon width 0.875 mm (Supplementary Fig. 13f and Supplementary note 6), arising from the same grasping transition mode from encapsulating to pinching (Supplementary Fig. 13a–e).

It is noteworthy that using pinching or the friction force, existing kirigami grippers are not well suited for grasping gelatinous organisms[17,48]. These grippers need to compress or pinch the targets to lift the targets, making the noninvasive collection of the delicate organisms challenging[17,48]. Distinct from that, we demonstrated a grasping mode, encapsulating the targets ultra-gently without compressing the objects, via the programmable dynamic morphology, which is especially suitable for grasping delicate organisms nondestructively. Also, the pulling-out force of our gripper could be an order of magnitude (about ten times) larger than recently reported kirigami grippers[48] at the same scale by harnessing the dynamically programmed morphology.

**Application: conformable heater composed of the geodesic ribbons.** Conformable heating devices are desired for human joints to relieve pain[50]. The inhomogeneous deformation and complex-curved shapes[51] of the joints, such as the human knee, result in the trade-off between the large contact area (between the device and the human skin) and the conformability and adaptivity, especially during motion. Here, different from existing heaters[32,52,53], we harness the correlation between the boundary

curvature and the 3D morphology and demonstrate an electrically driven resistive human-knee heater mimicking the conformability of the Mantis shrimp's shell. It shows intrinsic adaptivity with decent conformability and large-area uniform-heating capability.

As shown in Fig. 8a, the heater is composed of silver nanowires (AgNWs)[54,55] and the PDMS (polydimethylsiloxane) kirigami sheet with parallel cuts. It generates Joule heating with a constant direct current applied (see "Methods" section). Mimicking the curling shell of the Mantis shrimp (Fig. 8b), the heater includes discrete ribbons that are consistent with the geodesics of the knee, which are normal to the boundary. Four vertices of the kirigami AgNW-PDMS pad are bonded to the knee, as shown in Fig. 8c. As the knee bends from 0 to 90°, the discrete ribbons deform induced by the variation of the boundary curvature like the curling of the shell of the shrimp (Fig. 8d, e). The geodesic feature, same as the shell, endows intrinsic adaptivity, and the heater shows decent conformability and uniform-heating capability, where the standard deviation of the temperature across the knee before and after bending is 0.73 °C and 0.97 °C, respectively (Fig. 8d, e). With the increase of the bending angle $\varphi$, the temperature at the center of the heater (point M in Fig. 8d) slightly decreases, resulting from the small increase in the resistance of AgNWs (~1%), which shows stable heating capability (Fig. 8f). We note that the cyclic heating and cooling do not degrade the performance of the device, where the resistance-temperature curves barely change after 100 cycles of heating and cooling from 25 to 42 °C (Supplementary Fig. 14). The discrete ribbon-based kirigami design offers the unique features of adaptivity, conformability, and flexibility combined with a large contact area, which can be potentially applied to wearable sensors, flexible electronics, and textile electronics.

## Discussion

We proposed a new way of utilizing the cut boundary curvature to guide the formation of controllable and reconfigurable complex 3D-curved shapes in kirigami sheets patterned with simple parallel cuts. Such a strategy is validated through combined theoretical modeling, FEM simulations, and experiments. The unique feature of discrete cut ribbons as geodesic curves of the deformed 3D shapes largely simplifies the inverse design. Programming the dynamic 3D morphology, we showed a universal

noninvasive flexible kirigami gripper for grasping and preserving delicate organisms and a biomimetic kirigami heater with decent conformability and intrinsic adaptivity to human knees.

The parallel cuts ensure easy fabrication. However, there are some limitations on our proposed method in terms of the achievable morphed shapes and the level of curvature programmability[56]. It is challenging for the straight discrete ribbons to approximate an axisymmetric shape perfectly due to the limitation of their elastica shape (e.g., a cone shape). The inverse design with a high-accuracy requirement will need local optimization of the boundary curve. For targeted more complex 3D surface shapes with arbitrary negative and positive curvatures, the inverse design will become more challenging since it needs to utilize the smoothness of two orthogonal geodesics to design both the tessellation of different shaped unit cells and the shapes of inner and outer boundary curves. Moreover, compared to the intrinsic deployment of retained 3D shapes through bistability[56] or pre-strain release[16] after force removal, our approach requires the application of external stretching forces to remain the deployed shapes, otherwise, the generated 3D shape will return to its original flat form after the external actuation is removed due to the fully reversible elastic deformation in the thermoplastic kirigami structure. To preserve the deformed 3D shapes, we could utilize the shape memory properties of the PET polymer upon heating above its glass transition temperature[57]. We use thermal treatment under 120 °C to treat the 3D shapes held at an applied stretching strain for a period of 120 min and cooled down to the room temperature to fix the deformed configuration (Supplementary Fig. 15b). Notably, the preserved 3D configuration can be further deformed and recover to its 2D flat precursor shape upon another thermal treatment (Supplementary Fig. 15c).

Despite the demonstration of programmable shaft shifting in the thermoplastic kirigami sheets, we envision that the proposed strategy is material and scale independent. We note that despite the large applied stretching strain $\varepsilon$, the maximum principal strain $\varepsilon_{\max}$ in the buckled ribbons with thickness of 127 μm remains small ($\varepsilon_{\max} < 1\%$ for $\varepsilon > 50\%$, Supplementary Fig. 16 and Supplementary Note 7), e.g., $\varepsilon_{\max} = 0.4\%$ in the deformed spherical shape at $\varepsilon = 30\%$ (Fig. 2d) and $\varepsilon_{\max} = 0.6\%$ in the saddle shape at $\varepsilon = 147\%$ (Fig. 2h). Note that at the tip of the cuts, the stress concentration could be reduced via curved cuts, and moderate plastic deformation could be tolerated, as demonstrated by the over 1400 repeated cycles of 400 g deadweight lifting with the gripper without failure. Considering the small peak tensile strain in the buckled ribbons and its linear relationship with sheet thickness $t$, i.e., $\varepsilon_{\max}$ decreases linearly with $t$, we envision that the proposed kirigami strategy could also be applied to design shape-morphing and stretchable structures and devices made of other functional materials such as metals and even semiconductors at small scales, as well as other stimuli-responsive materials actuated by temperature, electrical, and magnetic field, etc.

Besides the simple parallel cut pattern, we further explored applying the strategy of boundary curvature guided shape morphing in kirigami sheets to other homogeneous cut patterning, such as the triangular cut pattern. As demonstrated by the proof-of-concept experiment in Supplementary Fig. 17, it shows the formation of approximately similar curved surfaces as the parallel cuts by manipulating different boundary curvatures of the 2D precursors, but arising from distinct both local and global out-of-plane buckling in the cut units (Supplementary Note 8). The detailed deformation mechanism and its potential generality to other cut patterns will be explored and examined in the future. This work could find potential applications in designing soft robots, non-invasive soft grippers, stretchable electronics, wearable devices, and portable, and wearable heaters.

## Methods

**Simplification of the Gauss–Bonnet theorem**. The Gauss–Bonnet theorem can be simplified according to the constant Euler characteristic and summation of the exterior angles. The Euler characteristic of the surface is a topological invariant and keeps a constant during deformation. For the surface formed by two neighboring discrete ribbons, as shown in Supplementary Fig. 1, the Euler characteristic is given by

$$\chi(\Omega) = V - E + F = 1, \tag{9}$$

where $V$, $E$, and $F$ denote the numbers of vertices, edges, and faces of the manifold $\Omega$, respectively. In this process, $V$, $E$, and $F$ do not change with the increasing strain (Supplementary Fig. 1c, f, i), resulting in a constant Euler characteristic.

The summation of the exterior angles does not change under tension and the variation in summation is expressed as

$$\Delta\left(\sum_{i=1}^{P} \theta_i\right) = 2\left\{\Delta\left[\alpha_{2s}\left(s_b + \frac{w_d}{\sin(\alpha_{2s})}\right)\right] - \Delta\left[\alpha_{2s}(s_b)\right]\right\} = 0, \tag{10}$$

where $\Delta(\sum_{i=1}^{P} \theta_i)$ denotes the variation of the summation of the exterior angles; $\alpha_{2s}(s_b)$ denotes the angle between the tangent line of the boundary ribbon and the discrete ribbon at the point of intersection and is a function of the arc length of the boundary ribbon $s_b$; the coefficient 2 is from the symmetry of the structure; $w_d$ is the width of the discrete ribbons. Note that we assume the distance $\triangle s_b$ between two discrete ribbons along the boundary ribbon is expressed as $\triangle s_b = \frac{w_d}{\sin(\alpha_{2s})}$ because $w_d \ll R$, with $R$ being the half-width of the 2D precursor.

For the cylindrical and spheroidal shape, it is obvious that $\triangle \alpha_{2s} = 0$ during the deformation due to the conformal mapping (Supplementary Fig. 1). For the saddle shape, while $\alpha_{2s}$ is changing due to the contact between the discrete ribbons, and the summation of the exterior angles keeps constant because the variation of the angles of two neighboring discrete ribbons has the same absolute value. It is noteworthy that the Gauss–Bonnet theorem is first applied to the surface formed by two neighboring discrete ribbons and then extended to the entire structure.

**Variation of the geodesic curvature**. The curve of the ribbon is parametrized by arc length as shown in Supplementary Fig. 2a, c, e; the origin is located at the midpoint of each ribbon; $s_b$ and $s_d$ denote the arc length coordinate of the boundary and discrete ribbons, respectively. The integral of geodesic curvature along the smooth boundary of the manifold (Supplementary Fig. 1c) is composed of two parts, i.e., $\int k_g ds = \int k_{gd} ds_d + \int k_{gb} ds_b$, where $k_{gd}$ and $k_{gb}$ denote the geodesic curvature along the discrete and boundary ribbons, respectively. The geodesic curvature of the discrete ribbons is equal to zero ($k_{gd} = 0$), because the discrete ribbons are geodesics of the morphed surface (normal vectors of discrete ribbons are normal to the tangent plane). As such, the integral is simplified as

$$\int k_g ds = \int k_{gb} ds, \tag{11}$$

where $ds$ denotes the line element along the boundary of the manifold formed by two neighboring discrete ribbons.

Further, the geodesic curvature of the boundary ribbons is equal to the projection of the curvature of the boundary ribbon on the tangent plane $T_p$ of the surface, as shown in Supplementary Fig. 3. According to the Meusnier theorem[37], the relationship between the curvature of the boundary ribbon and the geodesic curvature is given by

$$k_{gb} = \langle \mathbf{r_b''}, \mathbf{S} \rangle = k_b \sin\varphi, \tag{12}$$

where $k_b$ is the curvature along the boundary ribbon. $k_{gb}$ is the geodesic curvature along the boundary ribbon. Vector $\mathbf{r_b}$ denotes the boundary curve parameterized by arc length and $|\mathbf{r_b'}| = k_b$. The angle $\varphi$ (Supplementary Fig. 3) is given in Eq. (2) of Supplementary note 1.

**Fabrication, mechanical testing, and thermal treatment of the kirigami sheets**. We used polyethylene terephthalate (PET) sheets (Dupont Teijin Film, McMaster–Carr) with Young's modulus of 3.5 GPa, Poisson's ratio of 0.38, and thickness of 0.127 mm for the kirigami sheets. The samples with different cut patterns were cut out using a laser cutter (EPILOG LASER 40 W) with cut ribbon width of 1.5 mm in Fig. 1a, b and 0.75 mm in Fig. 1c. Uniaxial tensile tests were performed using Instron 5944 to characterize the force–displacement curves under a loading rate of 10 mm/min. Thermal treatment (120 °C for 120 min) of the deformed PET samples under a stretched state in an oven fixed the generated 3D shapes upon cooling to room temperature and force removal. To recover to the initial flat state, a second thermal treatment of the deployed 3D shapes was conducted under 120 °C for 120 min in the oven.

**Finite element simulation**. In the FEM simulation (Abaqus/Standard), the PET sheets corresponding to three different morphologies as a spheroid, cylinder, and saddle were modeled as linear elastic, isotropic material with the measured Young's modulus of 3.5 GPa and Poisson's ratio of 0.38. The geometries were meshed with solid quadratic tetrahedral elements (C3D10H) and the fine mesh was applied to

the connection area for the ribbons. The left end was fixed and a prescribed displacement was applied to the right end to stretch the 2D kirigami precursors.

**Kirigami hand.** The kirigami hand is made of a polyethylene terephthalate (PET) sheet with patterned cuts via laser cutting. For a proof-of-concept demonstration, the shape-morphing of the gripper was manually actuated by uni-axial stretching. The two ends were attached to customized acrylic beams to perform tasks in the Supplementary Movie S3. The simplicity of the actuation makes it easy to be integrated with existing robotic platforms or to be actuated remotely via a magnetic field.

**Fabrication of the AgNW/PDMS heater.** For the nanowire synthesis, a modified polyol process was used. Firstly, 60 mL of a 0.147 M PVP (MW ~40,000, Sigma-Aldrich) solution in EG (was added to a round-bottom flask to which a stir bar was added; the vial was then suspended in an oil bath (temperature 151.5 °C) and heated for 1 h under magnetic stirring (150 rpm). Then at 1 h, 200 μL of a 24 M CuCl$_2$ (CuCl$_2$·2H$_2$O, 99.999+%, Sigma-Aldrich) solution in EG was injected into the PVP solution. The solution was then heated for an additional 15 min, followed by injecting 60 mL of a 0.094 M AgNO$_3$ (99+%, Sigma-Aldrich) solution in EG. AgNWs in ethanol solution with an average diameter of 90 nm and length of 20–30 μm were shaken for 5 min before use to disperse the nanowires in the solution. The AgNW solution was drop-casted on plasma-treated polyimide (PI) tape on a glass slide; at the same time, the solution was heated by a hot plate at 50 °C to evaporate the solvent. After the evaporation of ethanol, the AgNWs were thermally annealed at 150 °C for 20 min. Then the sample was laser cut to the desired pattern with extra nanowires and PI removed. Then liquid PDMS (SYLGARD 184) with a weight ratio of 10:1 was spin-coated onto the AgNW film, degassed, and subsequently thermally cured at 100 °C for 1 h. After curing, the AgNW/ PDMS composite was laser cut again to the designed geodesic pattern. Then the AgNW/PDMS composite was peeled off from glass/PI substrate in water. Finally, Cu wires were attached to the two ends of the heater by silver epoxy (MG Chemicals) for connection to the power source.

## Data availability

The authors declare that the data supporting the findings of this study are available within the article and its Supplemental Information files. Extra data are available from the author upon request.

## Code availability

The code used for the analyses will be made available upon e-mail request to the corresponding author.

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

## Acknowledgements

J.Y. acknowledges the funding support from the National Science Foundation under award number CMMI-2005374 and 2010717. We thank I. Kogan and Y. Zhao for helpful discussions.

## Author contributions

Y.H. and J.Y. designed research. Y.H. conducted the theoretical modeling and robotic demonstrations, Y.C. performed finite element simulation. S.W. and Y.Z. designed and conducted the heater demonstration, Y.C. and Y.L. conducted the experimental demonstration and mechanical testing. All authors analyzed data. Y.H. and J.Y. wrote the paper and all the co-authors revised the paper.

## Competing interests

The authors declare no competing interests.
