## [Peer Review File · Nature Communications]

REVIEWER COMMENTS

Reviewer #1 (Remarks to the Author):

The authors presented a clear way to achieve programmable shape-morphing kirigami sheets and the inverse design of the structures. Very interesting demos are included in the study as well. The finding can have an impact on shape-morphing design beyond kirigami. The paper is well-written. The reviewer supports its publication after the reviewer address the following comments.

1. It seems most cases shown in the current manuscript are driven by uni-axial loading. It is understandable that the authors may want to highlight the importance of the boundary curvature in the 2D design. But the external loading seems to be another important factor to control the shape-morphing. It will be better if the authors can add some discussions on the effect of loading.

2. Related to comment 1, uni-axial loading will lead to anisotropic structures. Will this method be able to achieve axisymmetric shapes, such as cones? The inverse design is very interesting and promising. But it may be helpful if the authors can discuss the available design spaces a little more.

3. The bistable kirigami structures are very interesting. How did the authors achieve the reconfiguration between the two states? It will be helpful to add a few discussions on this.

4. There are several units for the face patterns in Figure 4A. Will the face change its patterns under different loading rates?

5. The last demo shows the Multiphysics coupling of the kirigami structures, which is very nice. Will the cooling-heating cycles change the material properties of the structure?

6. Based on the formula, Eq. (1) is defined on a smooth surface in 3D space. For kirigami structures, there may be some hollow spaces after deformation. Will these discontinuous spaces affect the accuracy of the equation, such as the calculation of the curvature integration?

Reviewer #2 (Remarks to the Author):

Boundary curvature guided programmable shape-morphing kirigami sheets

By Hong et al.

In this work, the authors applied the Gauss-Bonnet theorem (that correlates the geodesic curvature along the boundary with the Gaussian) to kirigami for designing shapeshifters that can transform from 2D to 3D structures. They have developed a rigorous analytical model for predicting the targeted 3D shapes from 2D precursors and validated it with experiments and finite element simulations. They further, used a combinatorial design approach to realize more complex shapes. Finally, they have demonstrated the application of the proposed framework by developing a gripper for the manipulation of delicate objects and a conformal heater for human knees. Overall, this work presents a comprehensive investigation that spans from theory to applications and distinguishes itself from related works by focusing on the boundary rather than bulk for designing intricate functional kirigami structures. Therefore, it opens a new path towards the inverse design of these classes of mechanical metamaterials. Overall, I am in favor of the publication of this work in Nature Communications given its broad contribution to different fields including applied mathematics, solid mechanics, and soft robotics, its sound methodology, and extensive case studies and experimental demonstrations.

I have a couple of comments and questions as follows:

Choice of cut pattern: Is the fact that your kirigami sheet is composed of parallel discrete ribbons, a limitation of your approach? Does this imply that your approach is not applicable to other cut patterns such as “staggered linear” or “orthogonal square” cuts? As mentioned by authors “Cuts divide the original continuous thin sheets into discretized cut units but without sacrificing the structural integrity.” Does the selected cut pattern (parallel discrete ribbons) compromise the structural integrity of the system?

Mechanical response: A piece of information that is missing in this article is the mechanical response of kirigami structures with different boundaries. How does the force-displacement of the three basic structures shown in Figure 1 look like for samples with fixed height and area but with different boundaries?

Buckling behavior: Authors mentioned that “in contrast to simultaneous buckling in generating the spheroidal and cylindrical shapes we observe a sequential buckling during the formation of the saddle shape in experiments”. What is the physical origin of the localized buckling behavior in concave configuration? Does this behavior disappear for large radii of curvature? From the movies, it seems the structure is frustrated until a specific strain and relaxes to a uniform structure. Can one design the width of the ribbon in such a way that the deformation is frustration-free?

Limitations: It seems while the proposed method can produce a variety of global Gaussian curvatures, detailed programming of curvature still requires optimization of local cuts. For example,

achieving the level of curvature programmability that is stated in the following reference is beyond the capability of the proposed method. I suggest that the authors explain the limitation of their methodology with respect to the state-of-the-art inverse kirigami design such as the following reference.

Chen, T., Panetta, J., Schnaubelt, M., & Pauly, M. (2021). Bistable auxetic surface structures. *ACM Transactions on Graphics (TOG)*, 40(4), 1-9.

Minor edit:

I would replace “ancient” with “traditional” throughout the text. The word "ancient" points to very past time!

Reviewer #3 (Remarks to the Author):

In this paper, Hong et al. proposed a new approach for designing shape-morphing kirigami sheets by controlling the boundary curvature. Using the Gauss-Bonnet theorem in differential geometry, the authors connected 3D curved shapes with the geodesic curvature of the boundary curves. Both numerical simulations and physical experiments were presented to demonstrate the effectiveness of the approach. The authors also explored two applications of the approach for designing kirigami grippers and conformable heaters. Overall, this paper is interesting and well-written. I think the article is well-suited for *Nature Communications*.

I have a few questions and comments:

1. From a theoretical point of view, it is not entirely clear to me why setting the circular/square/biconcave boundary curves will lead to the spheroidal/cylindrical/saddle shapes as observed. With the increasing applied strain, surely the total Gaussian curvature $\int K \, dA$ will change correspondingly. However, why will K be globally positive/zero/negative? In other words, why won't there be some regions with a positive K and some regions with a negative K in the morphed shape? It will be great if the authors can discuss this in more detail.

2. By the Gauss-Bonnet theorem, the total Gaussian curvature is related to the surface topology. If one fixes the boundary curvature and introduces several holes in the kirigami sheet, will there be any changes in the morphed shape?

3. In the analysis of the relationship between the boundary curvature and the pulling-out force (Fig. 6J), there is a sharp transition at $k = 1$ but the force remains around the same from $k = 1$ to $k = 2$.

Does further increasing the boundary curvature lead to a larger F ? Is the performance also affected by the number of parallel cuts?

Reviewer #1:

Comments:

Summary: *“The authors presented a clear way to achieve programmable shape-morphing kirigami sheets and the inverse design of the structures. Very interesting demos are included in the study as well. The finding can have an impact on shape-morphing design beyond kirigami. The paper is well-written. The reviewer supports its publication after the reviewer address the following comments.”*

Response: We thank the reviewer for the positive comments on our work. The comments and suggestions have greatly helped us improve our manuscript. In the revised version, we have highlighted all the revisions in BLUE color.

Comment 1: *“1. It seems most cases shown in the current manuscript are driven by uni-axial loading. It is understandable that the authors may want to highlight the importance of the boundary curvature in the 2D design. But the external loading seems to be another important factor to control the shape-morphing. It will be better if the authors can add some discussions on the effect of loading.”*

Response: We agree with the reviewer that the external loading also plays an important role in the shape morphing. The boundary curvature determines the global shape, the increasing curvature of which will be controlled by the external loading. The effect of the applied strain on the curvature change can be seen in Eq. (2) on the modeling of shape functions. However, the loading will not transform its global shape from positive to negative Gaussian curvature, i.e., the sign of Gaussian curvature will remain the same with the increase of loading. Furthermore, we have also correlated the shape morphing and the force in terms of force-displacement curves.

In response to the reviewer’s comment, in the revised version, we have added the following in the main text: On Page 4,

“Once the 3D shape is formed, the global shape will not change but with its magnitude of curvature increasing with the applied strain. The three samples exhibit similar J-shaped force-displacement curves as shown in Fig. 1g, where the force increases approximately linearly with the initial displacement due to the bending-dominated deformation in the discrete ribbons, followed by the steep rise, arising from the stretching-dominated deformation in the boundary ribbon. Such

stiffness strengthening mechanical responses are similar to that observed in the kirigami sheet patterned with orthogonal square cuts²⁴. Among the three samples, the circular one morphing into a spheroidal shape shows the highest stiffness and the least stretchability, while the biconcave one deforming into a saddle shape is the most compliant and stretchable (Fig. 1g). ”

Comment 2: “2. Related to comment 1, uni-axial loading will lead to anisotropic structures. Will this method be able to achieve axisymmetric shapes, such as cones? The inverse design is very interesting and promising. But it may be helpful if the authors can discuss the available design spaces a little more.”

Response: We thank the reviewer for the good question.

Our method currently cannot achieve a perfect axisymmetric shape, such as cones. This is arising from the limitation of the elastica, which is the shape of the discrete ribbons, instead of the anisotropy induced by uni-axial loading. As shown in left figure, we use the elastica curve to approximate a unit circle. There always exists a small discrepancy. The elastica curve can mimic the shape of a circle but cannot form a perfect circle. In this work, we use the straight discrete ribbons, and we leave it as future work

to harness curved discrete ribbons allowing more complex morphologies.

Following the reviewer’s suggestion, in the revised version, we have added more discussions about the limitations of our method in the Discussion section.

In response to the reviewer’s comment, in the revised version, we have added the following discussions:

On Page 16-17, we added

“The parallel cuts ensure easy fabrication. However, there are some limitations on our proposed method in terms of the achievable morphed shapes and the level of curvature programmability⁵⁶. It is challenging for the straight discrete ribbons to approximate an axisymmetric shape perfectly due to the limitation of their elastica shape (e.g., a cone shape). The inverse design with a high-accuracy requirement will need local optimization of the boundary curve. For targeted more complex 3D surface shapes with arbitrary negative and positive curvatures, the inverse design will become more challenging since it needs to utilize the smoothness of two orthogonal geodesics to

design both the tessellation of different shaped unit cells and the shapes of inner and outer boundary curves. Moreover, compared to the intrinsic deployment of retained 3D shapes through bistability⁵⁶ or pre-strain release¹⁶ after force removal, our approach requires the application of external stretching forces to remain the deployed shapes, otherwise, the generated 3D shape will return to its original flat form after the external actuation is removed due to the fully reversible elastic deformation in the thermoplastic kirigami structure.”

Comment 3: “3. The bistable kirigami structures are very interesting. How did the authors achieve the reconfiguration between the two states? It will be helpful to add a few discussions on this.”

Response: We thank the reviewer for the comment. The bistable states of the ribbons could be either manually switched or potentially remotely tuned using the magnetic field (Supplementary Fig. 11 and Supplementary Video 2).

In the revised version, on Page 11, we added

“It is worth noting that the bistable states of the ribbons could be either manually switched²⁵ or potentially remotely tuned using the magnetic field (Supplementary Fig. 11, Supplementary Video 2 and Supplementary note 4).”

Comment 4: “4. There are several units for the face patterns in Figure 4A. Will the face change its patterns under different loading rates?”

Response: We thank the reviewer for the thoughtful comment. Following the reviewer’s comment, we have conducted uni-axial tensile tests under different loading rates. The 2D precursor exhibits the same 3D shape under different strain rates from $2.5 \cdot 10^{-4} \text{ s}^{-1}$ to $9.5 \cdot 10^{-2} \text{ s}^{-1}$. The faces in Figure 4A now Fig. 4b-d also preserve the same pattern under different loading rates.

In response to the reviewer comment, in the revised version, we have added

On Page 10,

“We note that the face will not change its pattern under different loading rates.”

Comment 5: “5. The last demo shows the Multiphysics coupling of the kirigami structures, which is very nice. Will the cooling-heating cycles change the material properties of the structure?”

Response: We thank the reviewer for the good question. Following the reviewer’s comment, we have conducted cooling-heating cyclic tests to demonstrate the change of the material properties, as shown in the figure below and Supplementary Fig. 14.

This figure shows the variation of the resistance of the heater as a function of the temperature. After 100 cycles of heating and cooling from 25°C to 42°C, the resistance-temperature curve almost does not change and the resistance is totally reversible. The maximum difference in the resistance between the two curves induced by the heating and cooling cycles is about 0.3%.

In response to the reviewer’s comment, in the revised version, we have added the following:

In the main text, on Page 16, we added:

“We note that the cyclic heating and cooling do not degrade the performance of the device, where the resistance-temperature curves barely change after 100 cycles of heating and cooling from 25°C to 42°C (Supplementary Fig. 14).”

Comment 6: “6. Based on the formula, Eq. (1) is defined on a smooth surface in 3D space. For kirigami structures, there may be some hollow spaces after deformation. Will these discontinuous spaces affect the accuracy of the equation, such as the calculation of the curvature integration?”

Response: We thank the reviewer for the comments. Eq. (1) is the motivation of our method, which is used only in the qualitative analysis of the formation of morphed shapes with different Gaussian curvatures, because the definite integral in the Gauss-Bonnet theorem prohibits the derivation of the Gaussian curvature at a specific point from the definite integral across the area. In the quantitative analysis, the Gaussian curvatures are calculated based on Eqs. (2)-(4). The hollow space between discrete ribbons has been considered in the model of the morphology.

Following the reviewer’s comment, in the revised version, we compare the constant C derived from differential geometry and C from the model based on solid mechanics. First, $C = 2\pi\chi(\Omega) - \sum_{i=1}^p \theta_i$, keeps zero during shape shifting due to the unchanged Euler characteristic and the summation of the exterior angles based on the Gauss-Bonnet theorem. $\chi(\Omega)$, θ_i , and p denote the Euler characteristic of the Riemannian manifold Ω with boundary $\partial\Omega$, the exterior angles at the vertices of the manifold, and the numbers of the vertices, respectively. Second, C can also be expressed as $C = \int_{\Omega} K dA + \int_{\partial\Omega} k_{gb} ds$, where $\int_{\Omega} K dA$ and $\int_{\partial\Omega} k_{gb} ds$ are calculated by Eqs. (2)-(4) based on the solid mechanics. K and k_{gb} denote the Gaussian curvature of the morphed surface and the geodesic curvature along the boundary ribbon.

(from solid mechanics) and orange (from differential geometry) curves are small compared with $\int_{\Omega} K dA$ and can be neglected.

To demonstrate the variation of C from these two equations during shape shifting, we use C as a function of the normalized geodesic curvature \bar{k}_{gb} along the boundary ribbon. As shown by the figure, $C = \int_{\Omega} K dA + \int_{\partial\Omega} k_{gb} ds$ is slightly larger than $C = 2\pi\chi(\Omega) - \sum_{i=1}^p \theta_i$ due to the hollow spaces. However, the discrepancies between the red

In the SI, we added the above discussions on Page 8 as below.

“**Analysis of the error induced by the holes.** First, $C = 2\pi\chi(\Omega) - \sum_{i=1}^p \theta_i$, keeps zero during shape shifting due to the unchanged Euler characteristic and the summation of the exterior angles based on the Gauss-Bonnet theorem. Second, C can also be expressed as $C = \int_{\Omega} K dA + \int_{\partial\Omega} k_{gb} ds$, where $\int_{\Omega} K dA$ and $\int_{\partial\Omega} k_{gb} ds$ are calculated by Eq. (2) in main text based on the solid mechanics. To demonstrate the variation of C from these two equations during shape shifting, we use C as a function of the normalized geodesic curvature \bar{k}_{gb} along the boundary ribbon. As shown in Supplementary Fig. 9, $C = \int_{\Omega} K dA + \int_{\partial\Omega} k_{gb} ds$ is slightly larger than $C = 2\pi\chi(\Omega) - \sum_{i=1}^p \theta_i$ due to the hollow spaces. However, the discrepancies between the red (from solid mechanics) and orange (from differential geometry) curves are small compared with $\int_{\Omega} K dA$ and can be neglected.”

Reviewer #2:

Comments:

Summary: *“In this work, the authors applied the Gauss-Bonnet theorem (that correlates the geodesic curvature along the boundary with the Gaussian) to kirigami for designing shapeshifters that can transform from 2D to 3D structures. They have developed a rigorous analytical model for predicting the targeted 3D shapes from 2D precursors and validated it with experiments and finite element simulations. They further, used a combinatorial design approach to realize more complex shapes. Finally, they have demonstrated the application of the proposed framework by developing a gripper for the manipulation of delicate objects and a conformal heater for human knees. Overall, this work presents a comprehensive investigation that spans from theory to applications and distinguishes itself from related works by focusing on the boundary rather than bulk for designing intricate functional kirigami structures. Therefore, it opens a new path towards the inverse design of these classes of mechanical metamaterials. Overall, I am in favor of the publication of this work in Nature Communications given its broad contribution to different fields including applied mathematics, solid mechanics, and soft robotics, its sound methodology, and extensive case studies and experimental demonstrations.”*

Response: We thank the reviewer for the positive comments on our work. The comments and suggestions have greatly helped us improve our manuscript. In the revised version, we have highlighted all the revisions in BLUE color.

Comment 1: *“Choice of cut pattern: Is the fact that your kirigami sheet is composed of parallel discrete ribbons, a limitation of your approach? Does this imply that your approach is not applicable to other cut patterns such as “staggered linear” or “orthogonal square” cuts? As mentioned by authors “Cuts divide the original continuous thin sheets into discretized cut units but without sacrificing the structural integrity.” Does the selected cut pattern (parallel discrete ribbons) compromise the structural integrity of the system?”*

Response: We thank the reviewer for the inspiring comments.

Following the reviewer’s suggestion, we have further examined the applicability of the boundary curvature guided shape morphing to kirigami sheets composed of other cut patterns. As a proof-of-concept experiment, we use the more complex triangular cut pattern with uniform patterning. The preliminary results show that the strategy does apply to the kirigami sheets with other cut patterns, as shown in the figure below. Starting from the planar triangular cut sheets with circular, rectangular, and biconcave boundaries with respective positive, zero, and negative boundary curvature, it forms approximately spheroidal, cylindrical, and saddle shapes upon stretching, respectively, which are similar to the case of parallel discrete ribbons.

Despite the similar shapes, there are several key distinctions in terms of surface smoothness and deformation mechanism:

(1) Different from the smooth discrete surface formed by parallel cuts, the generated discrete 3D surfaces are rather rough with extruded spikes, especially for the cases of spheroidal and saddle shapes with non-developable surface features (non-zero Gaussian curvatures). Both isometric and side-view images show that each triangular cut unit pops up and extrudes out of the curved surface.

(2) The deformation mechanisms for the formed shapes are different. For the triangular cuts, the deformation in each triangular cut unit is rather complex for the case of non-developable surfaces, because it undergoes both local and global out-of-plane buckling, such as pop up and rotation of cut units to open the cuts. Stretching leads to the non-uniform expansion and pore opening between the interconnected cut units. This is in contrast to the simple out-of-plane buckling of each discrete ribbon.

Thus, the strategy of boundary curvature-guided shape morphing is not limited to parallel cuts and can be applied to other different patterns but with different deformation mechanisms.

Regarding structural integrity, we meant that the compatible global deformation between discrete cut units. Thus, the parallel cuts will not compromise the global structural integrity in terms of coherent deformation.

In response to the reviewer's comment, in the revised version, we have added the following:

On Page 17 and 18, we added

“We envision that the strategy of boundary curvature guided shape morphing in kirigami sheets will not be limited to the simple parallel cut pattern. It could also apply to other homogeneous cut patterning such as the triangular cut pattern, as demonstrated by the proof-of-concept experiment in Supplementary Fig. 17. It shows the formation of similar curved surfaces as the parallel cuts by manipulating different boundary curvatures of the 2D precursors, but arising from distinct both local and global out-of-plane buckling in the cut units (Supplementary note 8). The detailed deformation mechanism and shape morphing will be explored in future.”

On Page 2, we modified as

“Cuts divide the original continuous thin sheets into discretized cut units but without sacrificing the **global** structural integrity.”

In the SI, we also added related discussions on Page 13 in “**Supplementary note 8**. Extending the strategy of boundary curvature guided shape morphing to other cut patterns.”

Comment 2: “*Mechanical response: A piece of information that is missing in this article is the mechanical response of kirigami structures with different boundaries. How does the force-displacement of the three basic structures shown in Figure 1 look like for samples with fixed height and area but with different boundaries?*”

Response: We thank the reviewer for the good comment. Following the reviewer's suggestion, we have conducted mechanical testing to characterize and evaluate the different mechanical responses of the kirigami structures. The results are shown below and the new Fig. 1g.

In response to the reviewer’s comment, in the revised version, we have added a new Fig. 1g, and the following:

On Page 4, we added

“The three samples exhibit similar J-shaped force-displacement curves as shown in Fig. 1g, where the force increases approximately linearly with the initial displacement due to the bending-dominated deformation in the discrete ribbons, followed by the steep rise, arising from the stretching-dominated deformation in the boundary ribbon. Such stiffness strengthening mechanical responses are similar to that observed in the kirigami sheet patterned with orthogonal square cuts²⁴. Among the three samples, the circular one morphing into a spheroidal shape shows the highest stiffness and the least stretchability, while the biconcave one deforming into a saddle shape is the most compliant and stretchable (Fig. 1g).”

stretchability, while the biconcave one deforming into a saddle shape is the most compliant and stretchable (Fig. 1g).”

Comment 3: “*Buckling behavior: Authors mentioned that “in contrast to simultaneous buckling in generating the spheroidal and cylindrical shapes we observe a sequential buckling during the formation of the saddle shape in experiments”. What is the physical origin of the localized buckling behavior in concave configuration? Does this behavior disappear for large radii of curvature? From the movies, it seems the structure is frustrated until a specific strain and relaxes to a uniform structure. Can one design the width of the ribbon in such a way that the deformation is frustration-free?*”

Response: We thank the reviewer for raising the concern. Please see below our responses.

“*What is the physical origin of the localized buckling behavior in concave configuration?*”

The physical origin of the sequential buckling in concave configuration can be explained from both the differential geometry and mechanics perspectives as below:

From the differential geometry perspective, as shown in Fig. 2f, the concave boundary ribbon is gradually straightened from the border to the center, accompanied by the sequential variation of

the curvature along the boundary ribbon with the increasing applied strain. The flattened regions of the boundary curve induce the corresponding discrete ribbons to pop up. This is consistent with the Gauss-Bonnet theorem, where the variation of the geodesic curvature induces the change in the Gaussian curvature. In contrast, for the spheroidal shape, the curvature variation along the circular boundary ribbon happens at the same time. For the cylindrical shape, there is no curvature variation in the boundary ribbon.

From the solid mechanics perspective, the sequential buckling is because of the coupling effect of the concave boundary and the different critical buckling forces of the discrete ribbons. Considering the force equilibrium of the boundary ribbon along the y -axis, the sequential process can be illustrated by a dimensionless variable $\delta = \frac{P \sin(2\beta)}{2 \sum_{i=1}^n P_C^i}$, where P and $\sum_{i=1}^n P_C^i$ denote the applied tensile force and the summation of the critical buckling force of discrete ribbons, respectively and $\beta = \sin^{-1}(\frac{g}{l_s})$. For a concave boundary, we have $\delta < 1$, leading to a sequential buckling. While for the circular and the rectangular boundaries, we have $\delta = 1$, which means that all the discrete ribbons pop up simultaneously and the sequential buckling behavior ends.

“Does this behavior disappear for large radii of curvature?”

The sequential behavior disappears for the large radius of curvature, because the 2D precursor is close to a rectangle. There is almost no sequential curvature change in the boundary ribbon during deformation. Also, with the large radius of curvature, the critical buckling forces of the discrete ribbons are similar, and the concave shape disappears, which prohibits the sequential behavior.

“Can one design the width of the ribbon in such a way that the deformation is frustration-free?”

For the structure frustration, we have done more experiments by decreasing the number of ribbons with the width of the ribbon unchanged to make the structure frustration-free, as shown by the figure below and Supplementary Fig. 7. The decrease of the width of the ribbon is limited due to the resolution of laser cutter.

The left Fig. a shows the schematics of reducing the number of ribbons in the 2D precursor. When the number of the ribbons is reduced from 84 to 36, the morphed shape becomes frustration-free, as shown by the isometric view of the morphed shape on the right.

In response to the reviewer’s comment, in the revised version, we have added a new Supplementary Fig. 7, and the following:

In the main text, on Page 8, we added

“The physical origin of the sequential buckling is due to the coupling effects of the concave boundary geometry and different critical buckling forces of the discrete ribbons (Supplementary note 1), where the curvature varies sequentially during deformation along the boundary ribbon from its two ends to the center (Fig. 2f). Such sequential buckling behavior disappears for the large radius of curvature since the 2D precursor is close to a rectangle shape. As the applied strain further increases, the discrete ribbons contact with each other, leading to structural frustration. Reducing the number of the ribbons facilitates a frustration-free structure without self-contact (Supplementary Fig. 7).”

In the SI, we added the above discussions on Page 5 and Page 6.

Comment 4: “*Limitations: It seems while the proposed method can produce a variety of global Gaussian curvatures, detailed programming of curvature still requires optimization of local cuts. For example, achieving the level of curvature programmability that is stated in the following reference is beyond the capability of the proposed method. I suggest that the authors explain the limitation of their methodology with respect to the state-of-the-art inverse kirigami design such as the following reference. Chen, T., Panetta, J., Schnaubelt, M., & Pauly, M. (2021). Bistable auxetic surface structures. ACM Transactions on Graphics (TOG), 40(4), 1-9.*”

Response: We thank the reviewer for the constructive comment and the reference. We agree with the reviewer that there are some limitations on our proposed method in terms of the achieved shapes and the level of curvature programmability.

Following the reviewer's suggestion, in the revised version, we have added more discussions about the limitations of our method in the Discussion section. The reference paper is added as Ref. 54.

In response to the reviewer's comment, in the revised version, we have added the following discussions:

On Page 16-17, we added

“The parallel cuts ensure easy fabrication. However, there are some limitations on our proposed method in terms of the achievable morphed shapes and the level of curvature programmability⁵⁶. It is challenging for the straight discrete ribbons to approximate an axisymmetric shape perfectly due to the limitation of their elastica shape (e.g., a cone shape). The inverse design with a high-accuracy requirement will need local optimization of the boundary curve. For targeted more complex 3D surface shapes with arbitrary negative and positive curvatures, the inverse design will become more challenging since it needs to utilize the smoothness of two orthogonal geodesics to design both the tessellation of different shaped unit cells and the shapes of inner and outer boundary curves. Moreover, compared to the intrinsic deployment of retained 3D shapes through bistability⁵⁶ or pre-strain release¹⁶ after force removal, our approach requires the application of external stretching forces to remain the deployed shapes, otherwise, the generated 3D shape will return to its original flat form after the external actuation is removed due to the fully reversible elastic deformation in the thermoplastic kirigami structure.”

Comment 5: *“Minor edit: I would replace “ancient” with “traditional” throughout the text. The word “ancient” points to very past time!”*

Response: Agree. In the revised version, we have replaced “ancient” with “traditional”.

Reviewer #3:

Comments:

Summary: *“In this paper, Hong et al. proposed a new approach for designing shape-morphing kirigami sheets by controlling the boundary curvature. Using the Gauss-Bonnet theorem in differential geometry, the authors connected 3D curved shapes with the geodesic curvature of the boundary curves. Both numerical simulations and physical experiments were presented to demonstrate the effectiveness of the approach. The authors also explored two applications of the approach for designing kirigami grippers and conformable heaters. Overall, this paper is interesting and well-written. I think the article is well-suited for Nature Communications.”*

Response: We thank the reviewer for the positive comments on our work. The following comments and suggestions have greatly helped us improve our manuscript. In the revised version, we have highlighted all the revisions in BLUE color.

Comment 1-1: *“1. From a theoretical point of view, it is not entirely clear to me why setting the circular/square/biconcave boundary curves will lead to the spheroidal/cylindrical/saddle shapes as observed. With the increasing applied strain, surely the total Gaussian curvature $\int K dA$ will change correspondingly. However, why will K be globally positive/zero/negative? In other words, why won't there be some regions with a positive K and some regions with a negative K in the morphed shape? It will be great if the authors can discuss this in more detail.”*

Response: We thank the reviewer for the good question. In the revised version, we have added illustrations about the local and global formation of the Gaussian curvature, as shown below.

(1) From the theoretical perspective, our method can be understood within the framework of a competition between the geodesic curvature k_{gb} (projection of the boundary curvature k_b) and the Gaussian curvature K .

Before clarifying it, we would like to introduce the motivation of our method, the Gauss-Bonnet theorem, $\int_{\Omega} K dA + \int_{\partial\Omega} k_{gb} ds = C$. C is a constant during shape shifting and is expressed as $C =$

$2\pi\chi(\Omega) - \sum_{i=1}^p \theta_i$. $\chi(\Omega)$ and θ_i denote the Euler characteristic of the Riemannian manifold Ω with boundary $\partial\Omega$ and the exterior angles at the vertices of the manifold, respectively.

Setting the circular, square, and biconcave boundary provides a stretching-tuned mechanism of variation in the boundary curvature k_b and geodesic curvature k_{gb} , which further changes the Gaussian curvature. With the parallel cuts enclosed by a curved boundary, these structures provide a distinct routine to locally relieve the variation in the boundary curvature through changing the Gaussian curvature. For 2D precursor with $k_b > 0$, we have $\int_{\partial\Omega_0} k_{gb} ds = C$ by setting $K = 0$ with Ω_0 denoting the manifold before deformation. After deformation, for the deformed manifold Ω' , we have $\int_{\Omega'} K dA = C - \int_{\partial\Omega'} k_{gb} ds = \int_{\partial\Omega_0} k_{gb} ds - \int_{\partial\Omega'} k_{gb} ds$ in terms of Eq. (1). With the increasing strain, both k_b and $\sin \varphi$ decrease, which results in a decreased geodesic curvature k_{gb} . Therefore, we have $\int_{\Omega'} K dA > 0$ in the demonstrated spheroidal shape in Fig. 1 with $K > 0$, where $\int_{\Omega'} K dA$, and K are simultaneously positive or negative in three characteristic precursors with continuous curvature boundaries (C^2 continuity). For 2D precursor with $k_b = 0$, $k_b = 0$ does not change during the deformation, which leads to a zero geodesic curvature. Thus, we have a cylindrical shape with $K = 0$ in Fig. 1. Similarly, for the 2D precursor with $k_b < 0$, as the strain increases, the absolute value of the boundary curvature $|k_b|$ becomes smaller and $\sin \varphi$ decreases, which results in an increased geodesic curvature. Thus, we have a generated saddle shape with $K < 0$, as shown in Fig. 1.

(2) For the spheroidal and saddle shapes, the Gaussian curvature is globally positive and negative, respectively, because they have C^2 continuous boundary curves, where C^2 continuity represents the curve, the first derivative, and the second derivative are continuous (i.e., continuous in curvature) [1]. Upon stretching, the boundary curvature k_b of all the points along the C^2 continuous boundary curve decreases or increases, leading to the globally positive or negative Gaussian curvature. In other words, the curvature-increase and the curvature-decrease regions do not exist at the same time in a C^2 continuous boundary under uniaxial tension.

Further, in the combinatorial design, tuning the smoothness of the boundary curve affords the structure a new mechanism of combining the positive and negative Gaussian curvatures. As shown in the schematics of the 2D precursors, the

global shape with positive and negative Gaussian curvatures is generated by combining two boundary curves with C^2 continuity. The two boundary curves are connected by a critical point M , where the neighboring region of the critical point is C^1 continuous. C^1 continuity represents the curve and the first derivative are continuous (i.e., continuous in tangent vector). Upon stretching, it generates a shape with both positive and negative Gaussian curvature, as validated by the vase shape in Fig. 4n and Fig. 5e.

In response to the reviewer's comment, in the revised version, we have added the following discussions:

On Page 5, we added

“For the 2D kirigami precursor with positive boundary curvature, i.e., $k_{bo} > 0$, we have $C = \oint_{\partial\Omega_o} k_{gb} ds$ by setting $K = 0$ with Ω_o denoting the manifold before deformation. After deformation, for the deformed manifold Ω' , we have $\int_{\Omega'} K dA = C - \oint_{\partial\Omega'} k_{gb} ds = \oint_{\partial\Omega_o} k_{gb} ds - \oint_{\partial\Omega'} k_{gb} ds$ in terms of Eq. (1). As the applied strain increases, both k_b and $\sin \varphi$ decrease, which results in a decreased geodesic curvature k_{gb} , and consequently $\int_{\Omega'} K dA > 0$. Given the C^2 continuous boundary curves in the three characteristic precursors (C^2 continuity means that both the first and second derivatives of the curves are continuous, i.e., continuous in curvature), both $\int_{\Omega'} K dA$ and K will be simultaneously positive or negative. Thus, we have a globally positive K in the deformed manifold Ω' , i.e., $K > 0$ in Ω' , which is consistent with the observed spheroidal shape in Fig. 1d. Similarly, for the 2D precursor with $k_{bo} < 0$, as the strain increases, the absolute value of the boundary curvature $|k_b|$ becomes smaller and $\sin \varphi$ decreases, which results in an increased geodesic curvature. Thus, we have the generated saddle shape with globally $K < 0$ in Fig. 1f.”

In the SI, we also added related discussions on Page 8 in “**Supplementary note 3.** Effect of the boundary curve smoothness in combinatorial designs.”

Comment 2: “2. By the Gauss-Bonnet theorem, the total Gaussian curvature is related to the surface topology. If one fixes the boundary curvature and introduces several holes in the kirigami sheet, will there be any changes in the morphed shape?”

Response: We thank the reviewer for the thoughtful comment. Following the reviewer’s suggestion, we have conducted the proof-of-concept experiments by introducing holes in the kirigami sheets with circular, rectangular, and biconcave boundaries. The results are shown in the figure below.

As shown in the figure above, the holes with a random distribution are introduced in the three types of 2D precursors, with the numbers of the holes being 1, 3, and 5. Upon stretching, these introduced holes do not significantly change the global morphology of spheroidal, cylindrical, and saddle shapes. Differently, the local shapes are changed. The discrete ribbons passing the holes keep straight during deformation and rotate with the bending of the boundary ribbon, where the rotation is defined by the angle $\varphi = \tan^{-1}\left(-\frac{2\sqrt{m^2-m^4}}{1-2m^2}\right) + \frac{\pi}{2}$ (supplementary Eq. (2)), with m being the elliptical modulus. These holes change the topology, related to the Euler characteristic χ but disobey the prerequisites for the Gauss-Bonnet theorem, where the boundary needs to be closed, simple, piece-wise regular curves [1]. The holes broaden the design space of our method, and we leave it as future work. The scale bar represents 10mm.

Comment 3-1: “3. In the analysis of the relationship between the boundary curvature and the pulling-out force (Fig. 6J), there is a sharp transition at $k = 1$ but the force remains around the same from $k = 1$ to $k = 2$. Does further increasing the boundary curvature lead to a larger F ?”

Response: We thank the reviewer for the good comment. Following the reviewer’s suggestion, we have conducted more experiments to illustrate the variation in the pulling-out force F of the

grippers with increasing initial-boundary curvature \bar{k}_{bo} , as shown in the figure below and Supplementary Fig. 12. We find that further increase of the boundary curvature beyond 2 does not lead to a larger F. Rather, they show similar pulling-out forces of around 6 N (see fig. b below), since all the samples show similar closed shapes (see fig. a below), which results in the same encapsulating grasping mode.

In response to the reviewer’s comment, in the revised version, we have added a new Supplementary Fig. 12 and the following discussions:

On Page 14, we added

“Further increasing the normalized boundary curvature beyond 2 does not lead to a higher pulling force (Supplementary Fig. 12b), since all the grippers share a similar closed shape under the same encapsulating grasping mode (Supplementary Fig. 12a), where further deformation is constrained by the contacted hemisphere petals.”

Comment 3-2: “Is the performance also affected by the number of parallel cuts?”

Response: We thank the reviewer for the comment. Following the reviewer’s comment, in the revised version, we have added a new Supplementary Fig. 13 to illustrate the effect of the number N_C of parallel cuts on the performance (the pulling-out force) of the gripper, as shown in the figure below and Fig. 13.

It shows the front and side views of the stretched grippers with different numbers N_c of parallel cuts at the maximum applied strain. The scale bar represents 10mm. For the grippers with small numbers N_c of cuts, as shown in Fig. a ($N_c = 5$), the variation of the boundary curve (yellow curves) is constrained by the large width w_a of the discrete ribbon resulting from the small number N_c of cuts, which further, prohibits the formation and the closure of the two encapsulating hemisphere petals. With increasing number N_c of cuts and decreasing ribbon width, the boundary curves are straightened gradually (Fig. b-c), with the constraint induced by the large ribbon width released. When $N_c = 37$ (Fig. c), the two grasping petals become hemispheres, and the formation of the central saddle shape leads to the closure of the two hemispheres. Note that in the original version, the sample with $N_c = 43$ is used. Upon the closure of the encapsulating hemispheres, the grasping mode transfers from pinching to encapsulating, which results in a sudden jump of the pulling-out force (Fig. f). It is also noteworthy that after the critical point ($N_c = 37$), increasing N_c (Fig. e) barely changes the final deformed shape of the gripper, and the pulling-out force remains the same (6 N).

In response to the reviewer’s comment, in the revised version, we have added a new supplementary fig. 13 and the following discussions:

On Page 15, we added

“Furthermore, for the kirigami grippers with the same size and geometry, when reducing the number of parallel cuts or equivalently increasing the ribbon width (Supplementary Figs. 13a-e), we observe the similar sudden jumping of the dramatically reduced pulling force at a critical ribbon width 0.875 mm (Supplementary Fig. 13f and Supplementary note 6), arising from the same grasping transition mode from encapsulating to pinching (Supplementary Figs. 13a-e).”

In the SI, we added the above discussions in section “**Supplementary note 6. Performance of the modified kirigami grippers**” on Page 12.

References

1. Spivak M., *A Comprehensive Introduction to Differential Geometry*. (Publish or Perish, Boston, 1970), vol. 1.

REVIEWERS' COMMENTS

Reviewer #1 (Remarks to the Author):

The authors did a very nice job and addressed all the comments from the reviewer. It is an important piece of work. The reviewer recommends its publication as it is.

Reviewer #2 (Remarks to the Author):

The revised manuscript is accompanied by additional results that improved the quality of the work.

The authors have addressed all my comments, and most of their responses are convincing. However, their claim that the proposed method applies to other cut patterns is not strongly supported by experimental evidence. The additional experiments on triangular cut patterns clearly show that the obtained results are not conclusive if we compare them to the results of the ribbon pattern. Therefore, the author should rephrase the following statement, and instead of a conclusive statement, it is better to express it as a point of discussion.

"Thus, the strategy of boundary curvature-guided shape morphing is not limited to parallel cuts and can be applied to other different patterns but with different deformation mechanisms."

Apart from this minor comment, I believe the present work is a comprehensive and high-quality study, and I recommend it for publication in Nature Communications.

Reviewer #3 (Remarks to the Author):

The authors have addressed all my comments in the revised manuscript. I am happy to recommend acceptance of the paper for publication in Nature Communications.

Reviewer #2:

Comments:

Comment: *“The revised manuscript is accompanied by additional results that improved the quality of the work. The authors have addressed all my comments, and most of their responses are convincing. However, their claim that the proposed method applies to other cut patterns is not strongly supported by experimental evidence. The additional experiments on triangular cut patterns clearly show that the obtained results are not conclusive if we compare them to the results of the ribbon pattern. Therefore, the author should rephrase the following statement, and instead of a conclusive statement, it is better to express it as a point of discussion. “Thus, the strategy of boundary curvature-guided shape morphing is not limited to parallel cuts and can be applied to other different patterns but with different deformation mechanisms.” Apart from this minor comment, I believe the present work is a comprehensive and high-quality study, and I recommend it for publication in Nature Communications.”*

Response: We thank the Reviewer for supporting our manuscript for publication. The quoted statement was originally from the response letter not in the main text.

In the revised version, in the main text, we have rephrased the related sentences as a point of discussion in “Discussion” section.

On Page 17, we revised as “**Besides the simple parallel cut pattern, we further explored applying** the strategy of boundary curvature guided shape morphing in kirigami sheets to other homogeneous cut patterning, such as the triangular cut pattern.”

On Page 18, we revised as “it shows the formation of **approximately** similar curved surfaces”
“The detailed deformation mechanism and **its potential generality to other cut patterns** will be explored **and examined** in the future.”